# Pathogenic LRRK2 control of primary cilia and Hedgehog signaling in neurons and astrocytes of mouse brain

**Shahzad S Khan[1,2], Yuriko Sobu[1,2], Herschel S Dhekne[1], Francesca Tonelli[2,3], Kerryn Berndsen[2,3], Dario R Alessi[2,3], Suzanne R Pfeffer[1,2]***

[1]Department of Biochemistry, Stanford University School of Medicine, Stanford, United States; [2]Aligning Science Across Parkinson's (ASAP) Collaborative Research Network, Chevy Chase, United States; [3]MRC Protein Phosphorylation and Ubiquitylation Unit, University of Dundee, Dundee, United Kingdom

**Abstract** Activating LRRK2 mutations cause Parkinson's disease, and pathogenic LRRK2 kinase interferes with ciliogenesis. Previously, we showed that cholinergic interneurons of the dorsal striatum lose their cilia in R1441C LRRK2 mutant mice (Dhekne et al., 2018). Here, we show that cilia loss is seen as early as 10 weeks of age in these mice and also in two other mouse strains carrying the most common human G2019S LRRK2 mutation. Loss of the PPM1H phosphatase that is specific for LRRK2-phosphorylated Rab GTPases yields the same cilia loss phenotype seen in mice expressing pathogenic LRRK2 kinase, strongly supporting a connection between Rab GTPase phosphorylation and cilia loss. Moreover, astrocytes throughout the striatum show a ciliation defect in all LRRK2 and PPM1H mutant models examined. Hedgehog signaling requires cilia, and loss of cilia in LRRK2 mutant rodents correlates with dysregulation of Hedgehog signaling as monitored by in situ hybridization of *Gli1* and *Gdnf* transcripts. Dopaminergic neurons of the substantia nigra secrete a Hedgehog signal that is sensed in the striatum to trigger neuroprotection; our data support a model in which LRRK2 and PPM1H mutant mice show altered responses to critical Hedgehog signals in the nigrostriatal pathway.

**\*For correspondence:**
pfeffer@stanford.edu

**Competing interest:** The authors declare that no competing interests exist.

## Introduction

Mutations in the kinase encoded by the LRRK2 gene represent the predominant cause of familial Parkinson's disease (PD), a neurodegenerative disorder that results in the loss of dopaminergic neurons in the substantia nigra pars compacta (*Poewe et al., 2017*; *Alessi and Sammler, 2018*). LRRK2 encodes a protein kinase and recent work has shown that a subset of Rab GTPases comprise its primary substrates (*Steger et al., 2016*; *Steger et al., 2017*). Pathogenic mutations that localize to LRRK2's ROC domain (e.g. R1441C) and kinase domain (e.g. G2019S) increase its kinase activity in cells (*West et al., 2005*; *Greggio et al., 2006*; *Jaleel et al., 2007*; *Ito et al., 2016*; *Steger et al., 2016*), and interactions with other proteins including Rab29 (*Kuwahara et al., 2016*; *Purlyte et al., 2018*; *Liu et al., 2018*; *Gomez et al., 2019*) and VPS35 can also activate LRRK2 (*Linhart et al., 2014*; *Mir et al., 2018*). Reversal of LRRK2 phosphorylation is mediated at least in part by the PPM1H phosphatase that was recently discovered to specifically reverse LRRK2 action on multiple Rab GTPases (*Berndsen et al., 2019*). In cell culture, loss of PPM1H in wild-type mouse embryonic fibroblast (MEF) cells phenocopies the loss of cilia seen upon expression of pathogenic LRRK2 (*Berndsen et al., 2019*).

Rab GTPases are master regulators of protein trafficking and carry out their roles by binding to specific partner proteins when the Rabs are GTP-bound (*Pfeffer, 2017*; *Pfeffer, 2018*). Phosphorylation of Rab proteins interferes with their abilities to be loaded with GTP by cognate guanine

nucleotide exchange factors, a prerequisite for their binding to partner effector proteins (*Steger et al., 2016*; *Steger et al., 2017*). This alone would interfere with normal Rab GTPase function. Strikingly, once phosphorylated, Rab GTPases switch their preference and bind to new sets of phospho-specific protein effectors. For Rab8 and Rab10 these include RILPL1, RILPL2, JIP3, and JIP4 proteins (*Steger et al., 2017*; *Dhekne et al., 2018*; *Waschbüsch et al., 2020*) and Myosin Va (*Dhekne et al., 2021*). The consequences of new phospho-Rab interactions include pathways by which LRRK2 blocks ciliogenesis in cell culture and mouse brain via a process that requires RILPL1 and Rab10 proteins (*Steger et al., 2017*; *Dhekne et al., 2018*; *Sobu et al., 2021*); centriolar cohesion is also altered (*Madero-Pérez et al., 2018*; *Lara Ordónez et al., 2019*).

Although later stages of PD resemble those seen in Alzheimer's disease, PD is first and foremost a movement disorder, and studies to understand it's underlying causes must focus on understanding why PD is specifically characterized by dopaminergic neuron loss in the substantia nigra. LRRK2 is most highly expressed in immune cells, lung, kidney, and intestine, but it is also present in varying levels throughout the brain (*Lis et al., 2018*; *West et al., 2014*; *Mandemakers et al., 2012*). The striatum is comprised primarily of medium spiny neurons, interneurons and glial cells such as astrocytes. In an important study, *Gonzalez-Reyes et al., 2012* showed that dopaminergic neurons in the substantia nigra secrete Sonic Hedgehog (Hh) that is sensed by poorly abundant, cholinergic interneurons in the striatum. Hh is needed for the survival of both these cholinergic target cells and the Hh-producing dopaminergic neurons, despite the fact that only the cholinergic neurons express the PTCH1 Hh receptor. *Gonzalez-Reyes et al., 2012* showed further that Hh triggers secretion of glial derived neurotrophic factor (GDNF) from the cholinergic neurons, which provides reciprocal neuroprotection for the dopaminergic neurons of the substantia nigra.

We found previously that the rare, striatal, cholinergic interneurons that would normally sense Hh via their primary cilia are less ciliated in mice carrying the R1441C LRRK2 mutation (*Dhekne et al., 2018*). In that study, we proposed that cilia loss would decrease the ability of these cells to sense Hh signals. We show here that Hh signaling is indeed impacted by cilia loss in multiple LRRK2 mutant mouse models. Moreover, we show that striatal astrocytes share a broad ciliary deficit that likely impacts synaptic function.

## Results

### Primary cilia defects in striatal cholinergic interneurons of G2019S LRRK2 mice

We showed previously that cholinergic interneurons that represent about 5 % of the neurons in the dorsal striatum of 7 month, R1441C LRRK2 knock-in (KI) mice have fewer primary cilia than their wild type littermates (*Dhekne et al., 2018*). G2019S LRRK2 is the most common PD-associated mutation in humans, thus it was also important to investigate ciliation in the brains of G2019S LRRK2 mice. G2019S LRRK2 is a hyperactive kinase but less active than R1441C LRRK2 when assayed in cell culture (cf. *Steger et al., 2016*). The age of the mice was also important: mouse cilia lengthen with age (*Arellano et al., 2012*) and LRRK2 mutations are incompletely penetrant, making age an important variable in disease onset (*Lee et al., 2017*; *Domingo and Klein, 2018*).

Two different mouse lines were examined: 13 month C57BL/6 J mice carrying a genetic knock-in of human LRRK2 G2019S (hereafter referred to as G2019S LRRK2 KI; *Steger et al., 2016*) and 10 month C57BL/6 J mice overexpressing a BAC transgene encoding human G2019S LRRK2 (hereafter referred to as G2019S LRRK2 BAC Tg; *Li et al., 2010*). Similar to what we observed previously for 7 month R1441C LRRK2 mice, 13 month G2019S LRRK2 KI mice also displayed a significant loss of primary cilia in choline acetyltransferase (ChAT) interneurons of the dorsal striatum (*Figure 1A and B*). As before, primary cilia loss was cell-type specific as there was no difference in ciliation or cilia length in the surrounding cells (primarily medium spiny neurons) between mutant and wild-type groups (*Figure 1— figure supplement 1*). For cholinergic neurons that retained cilia, we detected no significant difference in length between wild type and mutant groups (*Figure 1C*). Cilia length is important to assess as it is thought to reflect signaling capacity (*Guo et al., 2017*).

G2019S LRRK2 BAC Tg mice overexpress LRRK2 protein by approximately six-fold (*Li et al., 2010*) and might be expected to display a more severe phenotype than the G2019S LRRK2 KI mice. In the brains of the 10 month G2019S LRRK2 BAC Tg mice, we also detected a decrease in ciliated

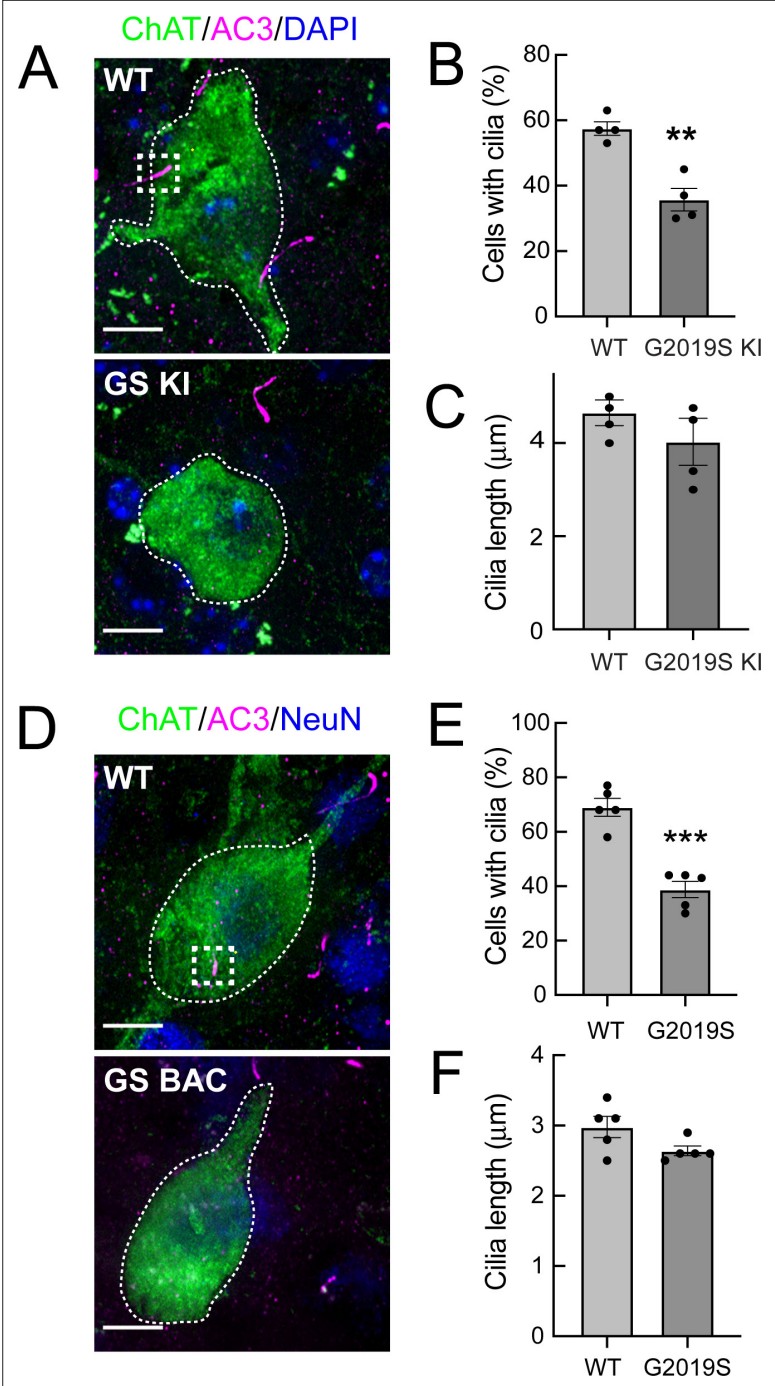

**Figure 1.** G2019S LRRK2 striatal cholinergic interneurons have fewer primary cilia. (**A**) Confocal images of sections of the dorsal striatum from 13 month wild-type (WT) or G2019S LRRK2 KI mice; Cholinergic Acetyltransferase (ChAT) (green, white outline); Adenylate cyclase 3 (AC3) (magenta, white box), and DAPI (blue). (**B**) Percentage of ChAT+ neurons containing a cilium. Wild type, light gray; G2019S KI, dark gray as indicated. (**C**) Quantitation of ChAT+ neuron ciliary length from sections as in A. (**D**) Confocal images of sections of the dorsal striatum of 10 month WT or G2019S LRRK2 BAC Tg mice; ChAT (green, white outline), AC3 (magenta, white box), and neuron specific nuclear antigen, NeuN (blue). (**E**) Percentage of G2019S LRRK2 BAC or wild-type ChAT+ interneurons containing a cilium. (**F**) Quantitation of ChAT+ neuron ciliary length. Scale bar, 10 µm. Significance was determined by t-test; (**B**) **, p = 0.0016; (**E**) ***, p = 0.0001. Values represent the data from individual brains, analyzing 4–5 brains per group, 2–3 sections per mouse, and >30 neurons per mouse.

The online version of this article includes the following figure supplement(s) for figure 1:

**Figure supplement 1.** Cilia density and cilia length in G2019S LRRK2 and wild type mice.

cholinergic interneurons (*Figure 1D and E*), comparable to that seen in the G2019S KI mice; when a primary cilium was present, it was not significantly shorter than wild type control cells (*Figure 1F*); again, there was no change in ciliation for the surrounding neurons. Taken together, these data show that G2019S LRRK2 mice have primary cilia defects of the same magnitude as previously observed in cholinergic neurons of the dorsal striatum of R1441C LRRK2 KI mice and cultured cells (*Steger et al., 2017*; *Dhekne et al., 2018*). Unlike the R1441C mice however, G2019S mice did not show defects in cortical ciliogenesis (*Figure 1—figure supplement 1*).

## Ciliary defects in G2019S striatal astrocytes

Mouse LRRK2 is more highly expressed in astrocytes than neurons (*Zhang et al., 2014*; http://www.brainrnaseq.org), thus it was important to explore the consequences of LRRK2 mutation on mouse astrocyte ciliation. We used glial fibrillary acidic protein (GFAP) and S100 Calcium Binding Protein B (S100B) to identify astrocytes in the striatum. Astrocytic primary cilia were detected using antibodies specific for ADP Ribosylation Factor Like GTPase 13B (Arl13B) because astrocytes in adult mouse brain do not express detectable amounts of Adenylate Cyclase three (AC3) that we use to stain cilia in neurons (*Dhekne et al., 2018*; *Sterpka and Chen, 2018*; *Sipos et al., 2018*; *Kasahara et al., 2014*).

Strikingly, we found that GFAP[+] G2019S LRRK2 KI and G2019S BAC Tg astrocytes in the dorsal striatum were less likely to have Arl13B[+] primary cilia relative to wild type controls (*Figure 2*). In addition, the remaining GFAP[+] cilia in the G2019S BAC Tg astrocytes were very slightly but significantly shorter (*Figure 2F*). Thus, GFAP[+] striatal astrocytes from G2019S LRRK2 KI mice have fewer primary cilia, and striatal astrocytes also have shorter primary cilia upon pathogenic LRRK2 overexpression. Note that astrocyte cilia were shorter overall compared with neuronal cilia (*Figures 1C, F, 2C and F*), and all cilia were shorter in the BAC Tg mice.

As shown in *Figure 3A and B*, primary cilia loss in cholinergic interneurons of the dorsal striatum could be detected as early as 10 weeks of age in R1441C LRRK2 KI mice; note that these were the youngest animals analyzed. At this age, fewer striatal cholinergic interneurons have a primary cilium (~40%) in comparison with their non-transgenic littermates (~60 %; *Figure 3B*). Overall neuronal ciliation in the dorsal striatum in 10 week wild type and R1441C mutant groups was slightly lower but similar to the overall values seen previously in a 7 month R1441C LRRK2 KI cohort (~70 %; *Dhekne et al., 2018*).

## MLi-2 treatment failed to reverse cilia loss in young R1441C LRRK2 KI Mice

LRRK2 kinase inhibitors are of great interest due to their potential to normalize LRRK2 kinase activity in patients carrying hyperactive LRRK2 mutant forms. Moreover, if ciliation is a relevant disease phenotype, it would be important to know if cilia defects can be corrected in mutant mice treated with a LRRK2 inhibitor. R1441C LRRK2 KI mice were fed the LRRK2 inhibitor, MLi-2, for 2 weeks for phenotypic analysis.

As shown in *Figure 3A, C and D*, 2 weeks of MLi-2 feeding did not alter the extent of ciliation or cilia length in cholinergic interneurons or astrocytes in the LRRK2 R1441C mice. We also scored primary cilia in two distinct populations of astrocytes – cells that were positive for both GFAP and S100B, and those that were only positive for S100B (*Figure 3E–G*). Despite 2 weeks of MLi-2 administration, we failed to detect a significant difference in ciliation or cilia length for either population of astrocytes upon treatment. It is important to note that neurons and astrocytes are postmitotic cells with much less dynamic cilia than dividing cells (*Sterpka and Chen, 2018*); longer times of drug treatment may very well be needed to reveal an effect.

The ability of MLi-2 to block LRRK2-mediated Rab10 phosphorylation in whole, R1441C LRRK2 mutant mouse brains was assessed by western blot (*Figure 4A and B*). LRRK2 inhibitors decrease LRRK2 phosphorylation at serine 935 (cf. *Fell et al., 2015*), and pS935-LRRK2 was diminished in wild-type and R1441C LRRK2 brains from mice fed MLi-2 inhibitor, consistent with inhibitor access in this tissue. Rab12 S105 phosphorylation also decreased in wild type (*Figure 4C*) and mutant (*Figure 4A and B*) brain. However, levels of pRab10 seemed unchanged (*Figure 4A and B*) with variability among animals (see also *Iannotta et al., 2020*). Previous studies have also noted that MLi-2 administration does not markedly reduce pRab10 levels in brain (*Kelly et al., 2018*; *Kalogeropulou et al., 2020*; *Nirujogi et al., 2021*; *Kluss et al., 2021*). Note that analysis of whole brain in this experiment will not

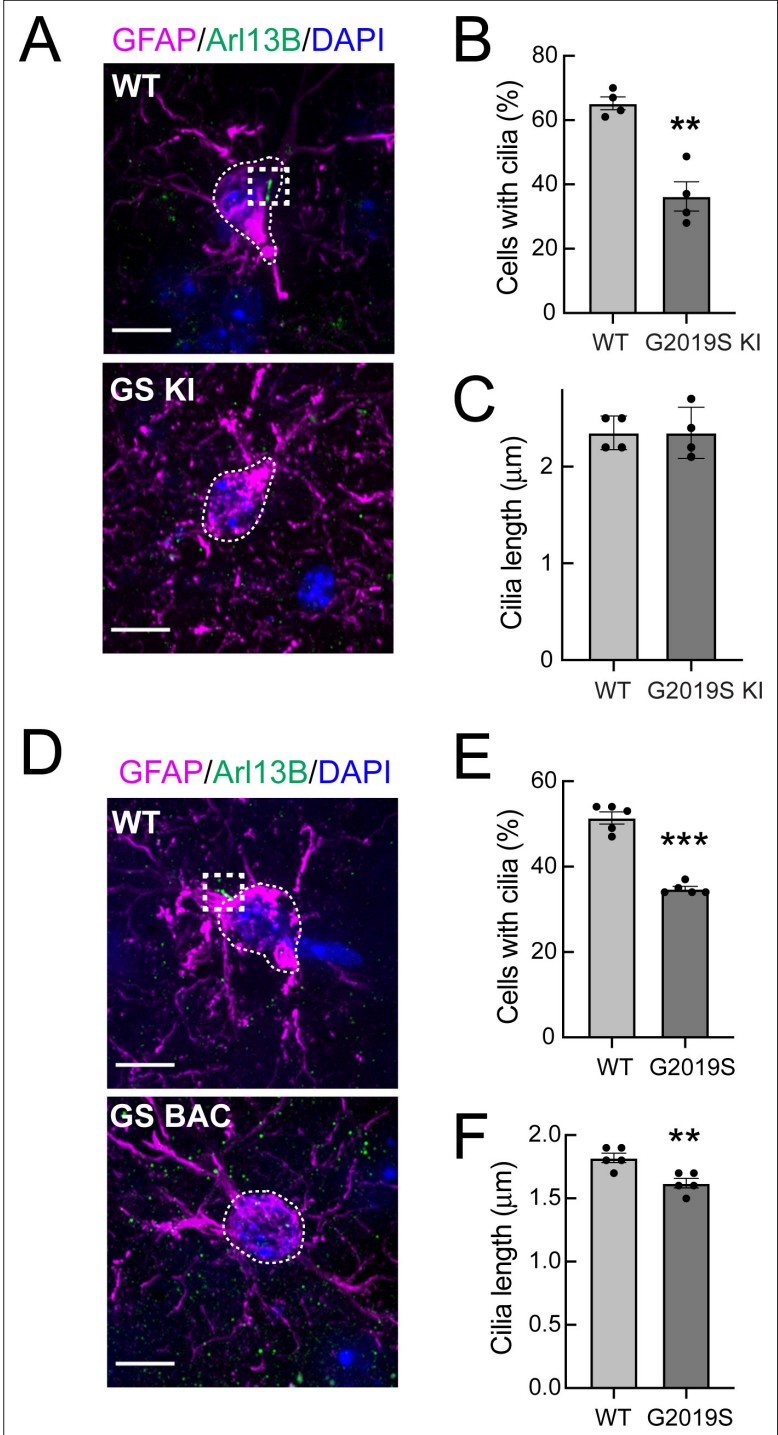

**Figure 2.** G2019S LRRK2 striatal astrocytes have fewer primary cilia. (**A**) Confocal images of sections of the dorsal striatum from 13 month WT or G2019S LRRK2 KI mice; astrocyte marker, Glial fibrillary acidic protein (GFAP) (magenta, white outline), cilia marker, ADP-ribosylation factor-like protein 13B (Arl13B) (green, white box), and DAPI (blue). (**B, C**) Quantitation of the percentage of astrocytes containing a cilium and astrocyte ciliary length from sections described in A. (**D**) Confocal images of sections of the dorsal striatum from 10 month G2019S LRRK2 BAC Tg mice, antibody labeled for Glial fibrillary acidic protein (GFAP, magenta, white box), Arl13B (green, white box), and stained with DAPI (blue). (**E, F**) Quantitation of the percentage of astrocytes containing a cilium and astrocyte ciliary length from sections described in D. Scale bars, 10 µm. Values represent 4–5 brains per group, 2–3 sections per mouse, and >30 astrocytes per mouse. Significance was determined by t-test; B, **, p = 0.0011; E, ** p = 0.0054; F, **** p < 0.0001.

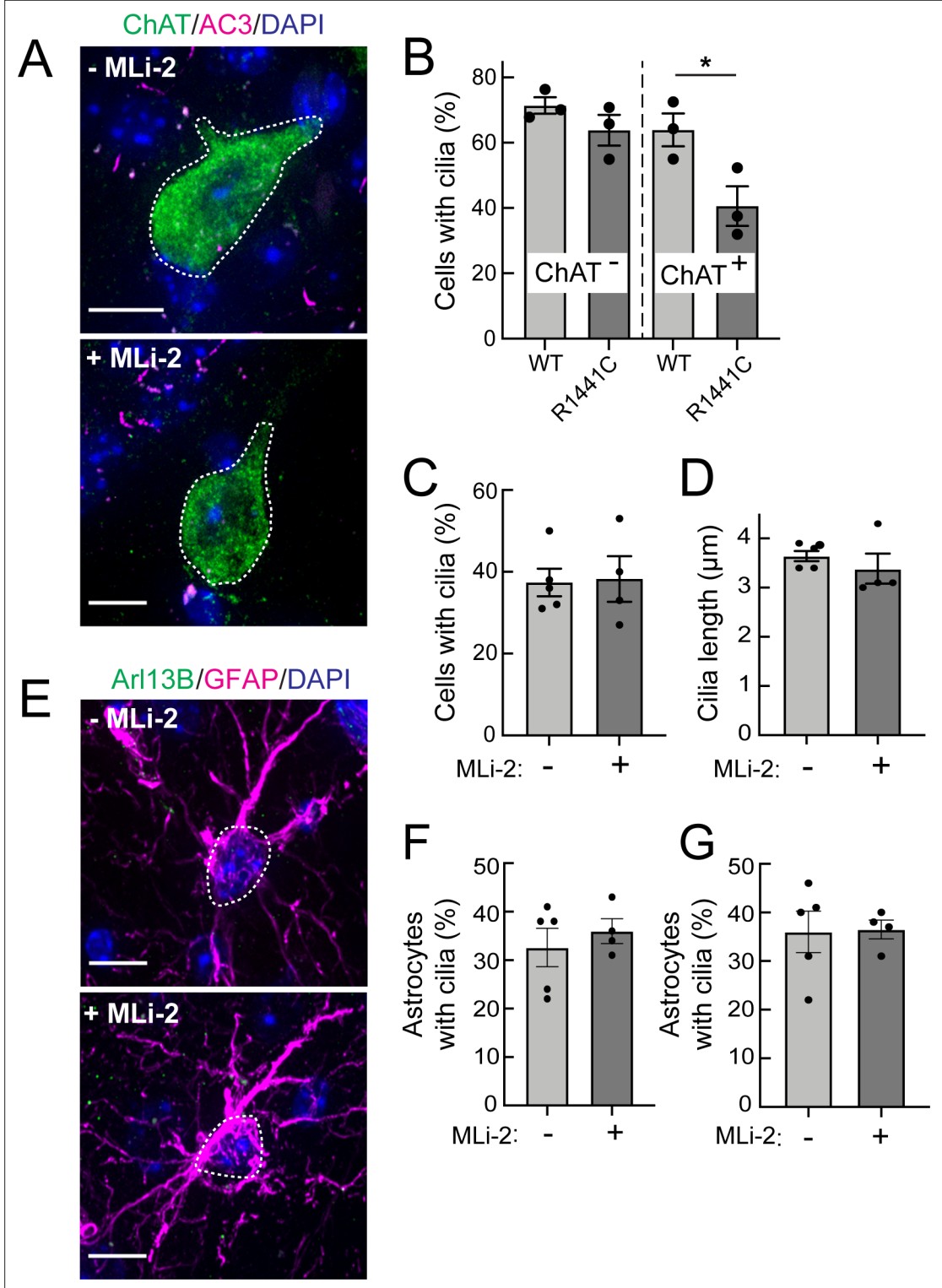

**Figure 3.** Two weeks MLi-2 treatment does not alter ciliogenesis in R1441C LRRK2 striatal interneurons or astrocytes. Mice (8 weeks old) were fed MLi-2 LRRK2 inhibitor-containing chow or control chow for 2 consecutive weeks prior to perfusion and staining. (**A**) Confocal images of sections of the dorsal striatum from 8 week R1441C LRRK2 KI mice; ChAT (green, white outline); Adenylate cyclase 3 (AC3) (magenta), and DAPI (blue). (**B**) Quantitation of the percentage of ChAT+ and ChAT- neurons containing a cilium. (**C**) Percentage of ChAT+ neurons containing a cilium± MLi-2. (**D**) Quantitation of ChAT+ neuron ciliary length. (**E**) Confocal images of astrocytes identified by antibodies to GFAP (magenta, white outline), Arl13B (green), and DAPI (blue), ± MLi-2. (**F, G**) Quantitation of the percentage of total astrocytes (**F**) or GFAP/S100B+ astrocytes (**G**) containing a cilium. Scale bars, 10 µm. Significance was determined by t-test. B, *, p = 0.0417.

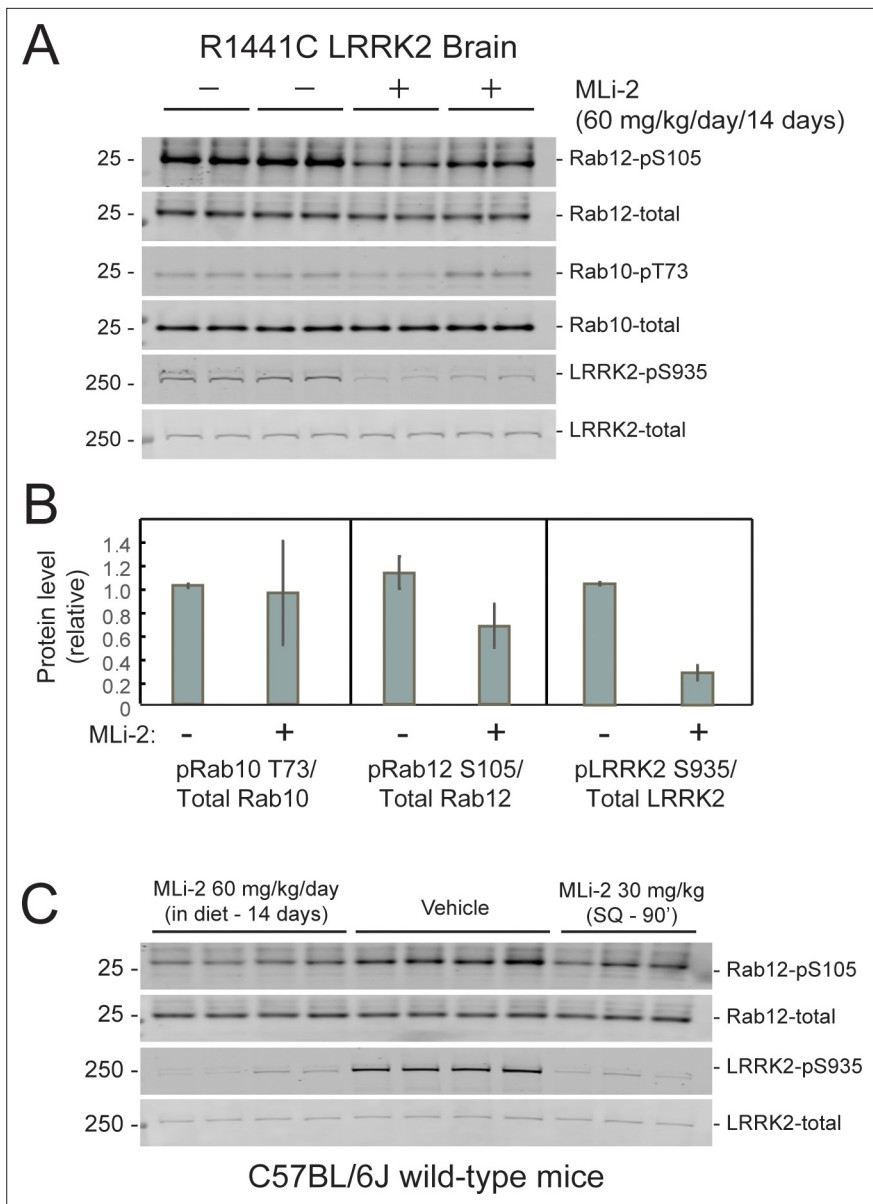

**Figure 4.** Two weeks of MLi-2 treatment decreases LRRK2 pS935 but does not alter Rab10 phosphorylation in R1441C LRRK2 mouse brain. (**A** and **B**). Littermate- or age-matched LRRK2 R1441C homozygous knock-in mice were fed either a control diet or MLi-2-containing diet for 14 days prior to tissue collection. Two mice from each group were perfused with PBS only; brains collected from these mice were snap frozen in liquid nitrogen and used to monitor inhibition of LRRK2 activity by immunoblotting. (**A**). 40 µg brain tissue extract was subjected to immunoblot analysis with antibodies specific for the indicated antigens. Duplicate samples were analyzed using the LI-COR Odyssey CLx imaging system. (**B**). Quantitation of data in A, calculated using Image Studio software (mean ± SD, normalized to control diet-fed animals.) (**C**). 11 C57BL/6 j wild-type mice received either control diet or diet containing MLi-2, targeted to provide a concentration of 60 mg/kg per day for 14 days. On the last day, 3 mice from the control diet group received 30 mg/kg MLi-2 dissolved in 40 % (w/v) (2-hydroxypropyl)-β-cyclodextrin via subcutaneous injection for 2 hr prior to tissue collection. Forty µg brain tissue extract was subjected to quantitative immunoblotting analysis with the indicated antibodies. Each lane represents a tissue sample from a different animal.

capture the precise status of pRab10 levels in rare cholinergic neurons of the dorsal striatum that show LRRK2 mutation-associated ciliary deficits. Unfortunately, the available anti-phosphoRab10 antibody cannot be used for tissue immunohistochemistry at a single-cell level.

## PPM1H deficiency phenocopies LRRK2 mutation

PPM1H phosphatase specifically dephosphorylates LRRK2 Rab GTPase substrates (*Berndsen et al., 2019*). If LRRK2 action is responsible for loss of cilia in mutant mouse brains, loss of the corresponding phosphatase should yield the same phenotype in that tissue. *Figure 5* shows that even heterozygous loss of PPM1H leads to decreased cilia numbers in cholinergic interneurons (*Figure 5A and B*) and astrocytes (*Figure 5C and D*) of the dorsal striatum of 5.5 month-old mice. Analysis of 2.5 month-old homozygous PPM1H knockout mice showed a slightly greater defect, despite the fact that fewer cholinergic neurons were ciliated at this age (compare *Figure 5B* left and right panels). These data demonstrate the importance of high PPM1H levels in wild type brain to counteract LRRK2 action. More importantly, they strongly validate a link between LRRK2-Rab phosphorylation and ciliogenesis in specific brain cell types because loss of the phosphatase shows the same phenotype as the presence of a hyperactive LRRK2 mutation. Immunoblotting showed an appropriate reduction of PPMH protein levels in brain extracts from *Ppm1h*$^{-/+}$ and *Ppm1h*$^{-/-}$ mice, with levels of pRab10 barely changed at least for whole brain tissue from the heterozygous knockout animals with a greater increase for the *Ppm1h*$^{-/-}$ animals (*Figure 5—figure supplement 1*).

## Sonic Hedgehog signaling is altered in mutant LRRK2 mice

Although Hh signaling is critical during neuronal development, little is known about Hh signaling in the adult brain. As mentioned earlier, striatal ChAT$^+$ interneurons respond to Sonic hedgehog ligands that originate from dopaminergic neurons in the substantia nigra. Sonic hedgehog signaling requires cilia: the transmembrane transducer Smoothened must translocate to the primary cilium to initiate the signaling cascade that results in the expression of the transcription factor *Gli1*, a direct Hh target gene that serves as a widely-used metric for signaling strength (*Corbit et al., 2005*; *Rohatgi et al., 2007*). Thus, primary ciliogenesis is likely critical for proper Hh sensing. We thus examined Hh signaling in LRRK2 mutant animals with pathogenic LRRK2 mutations and associated ciliary defects.

*Gli1* transcripts were detected in brain slices using the RNAscope method of fluorescence in situ hybridization. *Figure 6A* shows detection of *Gli1* transcripts in wild type, ChAT$^+$ neurons, which appear as white dots; a negative control hybridization probe yielded no signal under parallel conditions (*Figure 6A*, bottom row). As expected, ciliated ChAT$^+$ neurons showed higher levels of *Gli1* transcripts compared with non-ciliated cells in both mutant and wild-type mice (*Figure 6D*). Ciliated cholinergic neurons also displayed the highest number of *Gli1*-positive dots compared with all other cell types in the striatum (monitored by DAPI staining) and compared with non-ciliated cholinergic neurons (*Figure 6C and D*). This matches well with prior estimates that expression of the Hh receptor PTCH1 is restricted to 6 % of total striatal neurons representing all cholinergic and fast spiking interneurons (*Gonzalez-Reyes et al., 2012*).

Importantly, the number of *Gli1* dots trended higher in ciliated, striatal cholinergic interneurons of R1441C LRRK2 mice (3–4 *Gli1*-positive dots) compared with wild-type mice (*Figure 6D and E*). Taken together, these data show that although overall ciliation is decreased, the remaining, ciliated, striatal cholinergic interneurons in R1441C LRRK2 mice transcribe higher quantities of *Gli1* in response to R1441C LRRK2 expression. The upregulation of *Gli1* transcripts in the remaining ciliated neurons could be due to increased Hh production in the substantia nigra; perhaps the overall decrease in ciliation decreases GDNF production in the striatum, increasing Hh production by subsequently stressed, dopaminergic neurons. Alternatively, the remaining cilia in LRRK2 mutant brains may be structurally intact but functionally altered, leading to increased *Gli1* transcript levels. Finally, these data also reveal highly localized Hh signaling in cholinergic neurons in the dorsal striatum compared with their surrounding neighbors, as predicted by the distribution of PTCH1 protein (*Gonzalez-Reyes et al., 2012*).

Because R1441C LRRK2 striatal astrocytes also have fewer cilia, we determined if their ability to sense or respond to Hh was also altered. Astrocytes express higher levels of PTCH1 receptor than neurons in both rodent and human brain, and thus would be expected to be capable of Hh signaling. *Figure 7A and B* shows astrocytic *Gli1* detection using RNAscope; two classes of astrocytes were

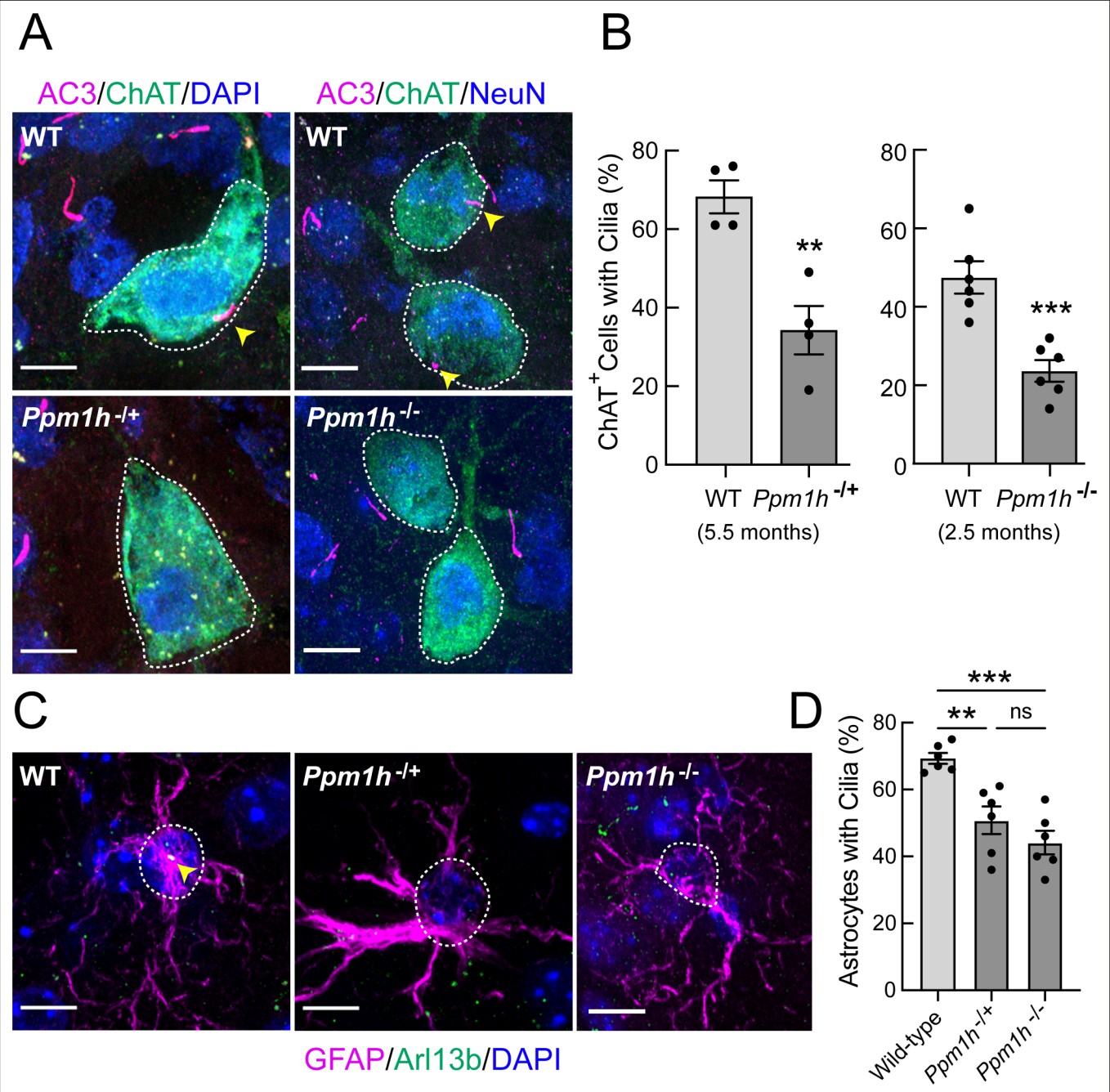

**Figure 5.** PPM1H mutant striatal cholinergic neurons and astrocytes have fewer primary cilia. (**A**) Confocal images of sections from 5.5 month *Ppm1h⁻/⁺* or 2.5 month *Ppm1h⁻/⁻* mice compared with corresponding age-matched WT mice; ChAT (green, white outline); AC3 (magenta, yellow arrowhead), and DAPI (blue). (**B**) Percentage of ChAT⁺ neurons of the indicated genotype containing a cilium. Wild type, light gray; *Ppm1h* mutant, dark gray as indicated. (**C**) Confocal images of sections of the dorsal striatum from mice described in A and B as indicated; GFAP (magenta), Arl13B (green, yellow arrowhead), and DAPI (blue). (**D**) Percentage of GFAP⁺ astrocytes containing a cilium. Wild-type, light gray; *Ppm1h⁻/⁺* or *Ppm1h⁻/⁻*, dark gray as indicated. Scale bars, 10 μm. Significance was determined by t-test; (**B**) WT vs *Ppm1h⁻/⁺*; **, p = 0.0018. (**C**) WT vs *Ppm1h⁻/⁻*; ***, p = 0.0007. (**D**) WT vs *Ppm1h⁻/⁺*; **, p = 0.0032. WT vs *Ppm1h⁻/⁻*; ***, p = 0.0002. *Ppm1h⁻/⁺* vs *Ppm1h⁻/⁻*; ns, p = 0.3474. Values represent the data from individual brains, analyzing four brains per group, 2–3 sections per mouse, and >30 cells per mouse. Significance was determined either by student's t-test or by Ordinary one-way ANOVA using Dunnett's multiple comparisons test. Scale bars, 10 μm.

The online version of this article includes the following figure supplement(s) for figure 5:

**Figure supplement 1.** Wildtype and PPM1H heterozygous or homozygous knock-out mice were treated with vehicle (40 % (w/v) (2-hydroxypropyl)-β-cyclodextrin) or 30 mg/kg MLi-2 dissolved in vehicle by subcutaneous injection 2 hr prior to tissue collection.

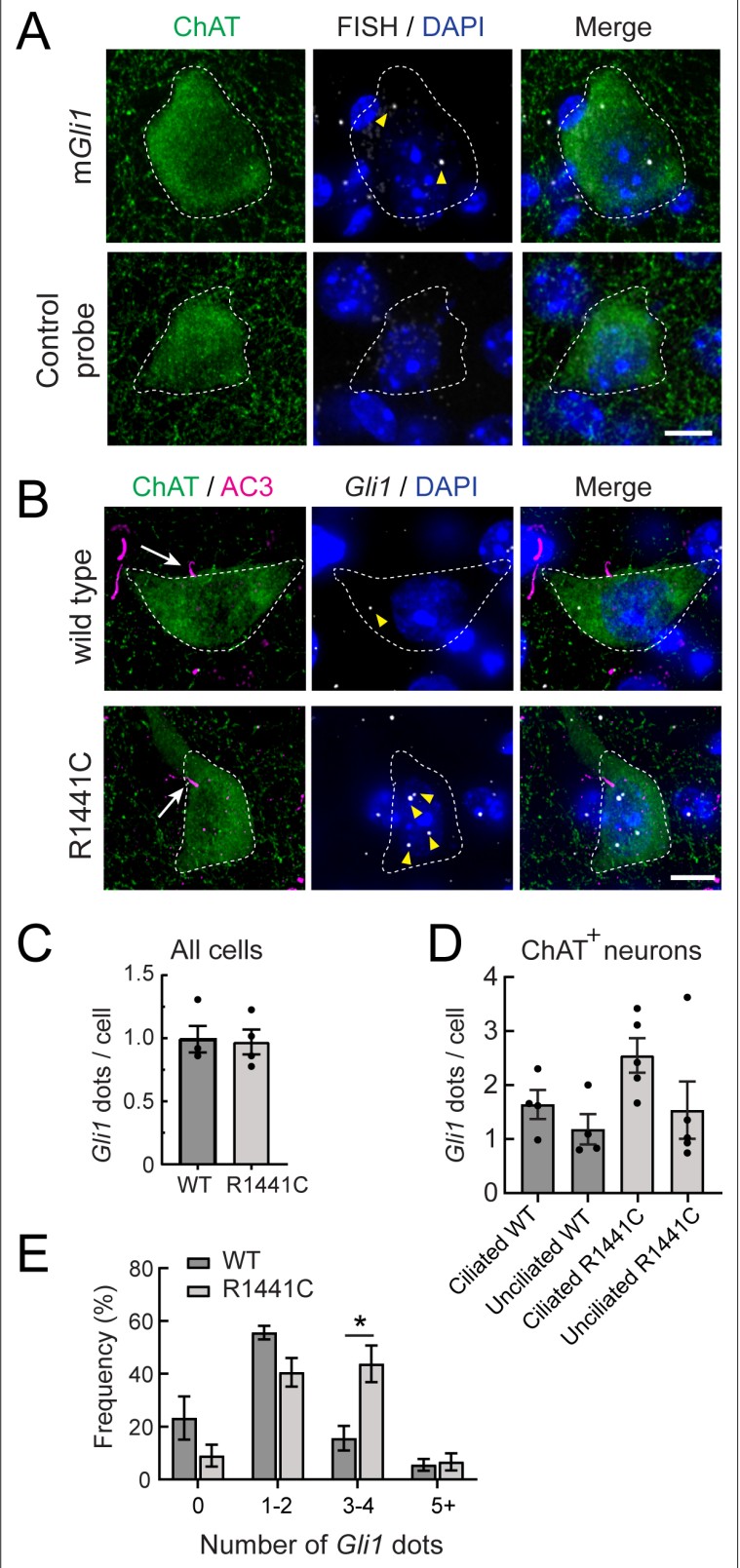

**Figure 6.** *Gli1* expression in R1441C LRRK2 dorsal striatal cholinergic Interneurons is cilia dependent and enhanced. (**A**) Ten  month WT mouse dorsal striatum was subjected to in situ hybridization using a *Gli1* probe (gray dots, highlighted by yellow arrowheads) or a negative control probe. ChAT (green, white outline) and DAPI+ nuclei (blue) were detected by immuno- or chemical staining. (**B**)  Ten month WT or R1441C mouse dorsal

*Figure 6 continued on next page*

*Figure 6 continued*

striatum was labeled as indicated: ChAT (green, white outline), AC3 (magenta, white arrow), *Gli1* mRNA (gray dots, yellow arrowheads), DAPI (blue). (**C**) Average numbers of *Gli1* dots per cell for all cell types in the dorsal striatum. Cell numbers were determined by DAPI staining. Values represent the mean ± SEM from 4 WT and 4 R1441C brains each containing >500 DAPI stained nuclei from 30 regions. p = 0.88. (**D**) Average numbers of *Gli1* dots for cholinergic interneurons with or without primary cilia as indicated. Values represent the mean ± SEM from 4 WT and 5 R1441C brains, each containing 9–32 cells. (**E**) Histogram of the number of *Gli1* dots in ciliated cholinergic interneurons from WT or R1441C mice. p = 0.14 (0), 0.054 (1-2),*,0.015 (3-4), 0.79 (5-). Significance was determined by t-test. Arrows indicate primary cilia for ChAT interneurons. Arrowheads indicate *Gli1* mRNA dots. Scale bars, 10 µm.

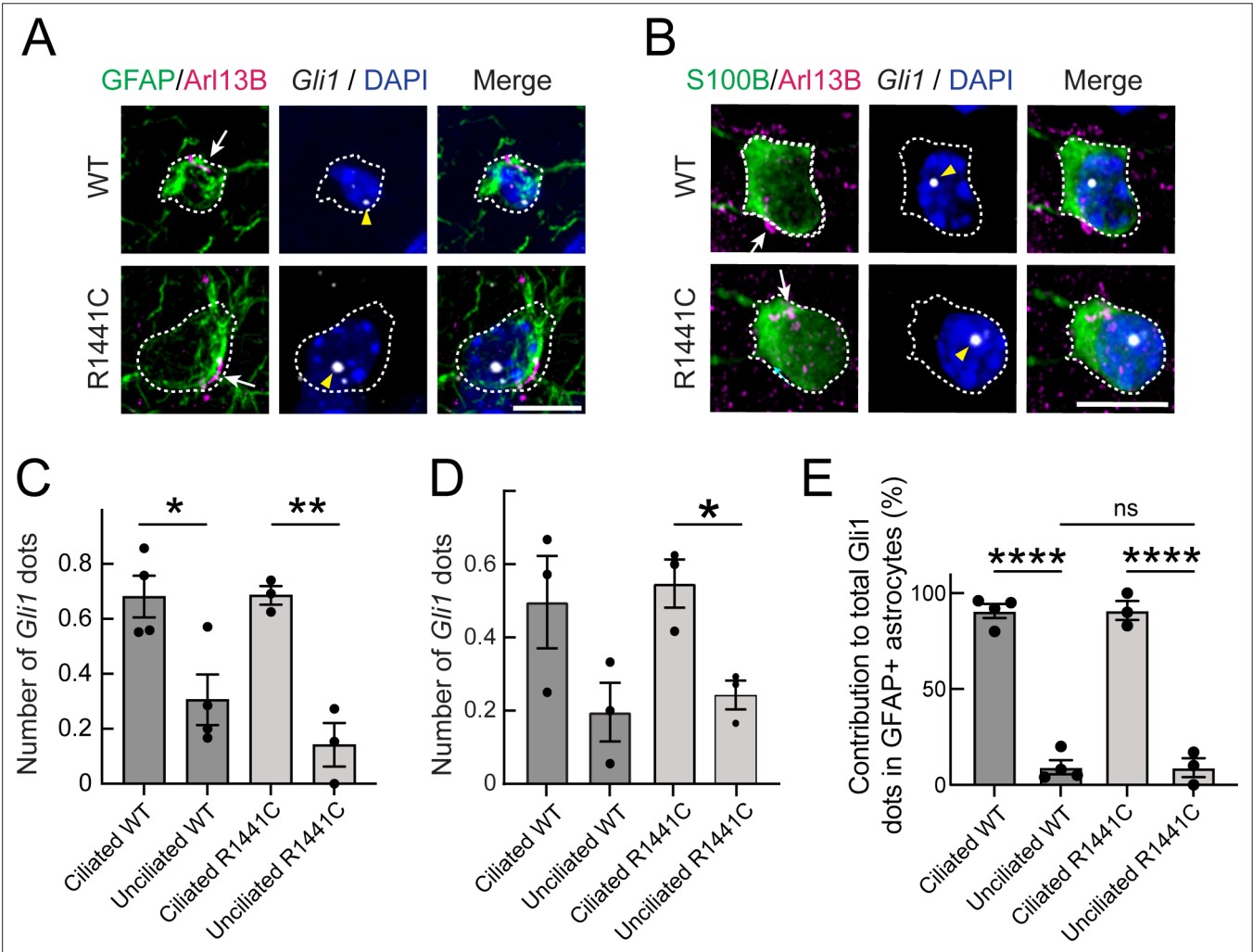

**Figure 7.** *Gli1* expression in astrocytes is cilia dependent. (**A,B**) Ten month R1441C LRRK2 KI mouse dorsal striatum was subjected to in situ hybridization using a *Gli1* probe (gray dots, highlighted with yellow arrowheads). Astrocytes were detected with (**A**) anti-GFAP or (**B**) anti-S100B (green, white outline); primary cilia were detected with anti-Arl13B (magenta, white arrows). (**C,D**) Average numbers of *Gli1* dots from (**C**) GFAP and (**D**) S100B+ astrocytes with or without primary cilia. (**E**) Relative contribution to total *Gli1* dots over GFAP+ astrocytes as a function of their ciliation status. Values represent the mean ± SEM from (**C**) 4 WT and 3 R1441C brains each containing >35 cells or (**D**) 3 WT and 3 R1441C brains each containing >21 cells. (**C**) Ciliated WT vs Unciliated WT: *, p = 0.020, Ciliated R1441C vs Ciliated R1441C: **, p = 0.0032. (**D**) Ciliated WT vs Unciliated WT: p = 0.12, Ciliated R1441C vs Unciliated R1441C: *, p = 0.017. (**E**) Ciliated WT vs Unciliated WT: ****, p < 0.0001; Ciliated R1441C vs Unciliated R1441C: ****, p < 0.0001; Unciliated WT vs Unciliated R1441C: ns, p > 0.9999. Significance was determined by unpaired t-test. Arrows indicate primary cilia from GFAP or S100B+ astrocytes. Scale bars, 10 µm.

scored: those expressing S100B or GFAP. Consistent with the requirement for cilia for canonical Hh signaling, ciliated astrocytes had more *Gli1* dots than non-ciliated astrocytes, independent of whether or not the cells expressed pathogenic R1441C LRRK2 (*Figure 7C and D*): in contrast to ChAT⁺ interneurons (*Figure 6*), ciliated and non-ciliated striatal astrocytes from R1441C LRRK2 had similar numbers of *Gli1* positive dots relative to their ciliated wild-type controls. Altogether, these data show that striatal astrocytes respond to Hh signaling similarly in R1441C LRRK2 and wild-type mice on an individual cell basis, but overall loss of ciliation due to the LRRK2 mutation appears to lead to an overall decreased striatal astrocytic response.

We next explored if pathogenic LRRK2-associated cilia deficits impact GDNF production by cholinergic neurons of the dorsal striatum. RNAscope fluorescence in situ hybridization for *Gdnf* transcripts showed that most GDNF was expressed by cholinergic neurons (and not surrounding cells) in the dorsal striatum of wild type mice, consistent with previous reports (*Hidalgo-Figueroa et al., 2012*; *Figure 8*). Moreover, *Gdnf* RNA was preferentially detected in ciliated wild-type cells, consistent with induction by a cilia-mediated pathway such as the Hedgehog pathway (*Figure 8A and D*). From this result, we anticipated that the increased proportion of non-ciliated cells in LRRK2 mutant animals would lead to an overall deficit in *Gdnf* transcription. Quantitation of total *Gdnf* transcripts failed to detect a significant decrease in overall *Gdnf* transcript signal in 5.5 month-old LRRK2 G2019S mice compared with wild-type animals, under conditions in which we detected an ~40 % loss of primary cilia (*Figure 8B,C*). *Gonzalez-Reyes et al., 2012* noted a strong age dependence for loss of GDNF production in the absence of Hh signaling; it is possible that older (12 month old) mice will show a greater defect in *Gdnf* transcript production. Also note that transcript levels may not correlate with protein production.

A completely unexpected result was obtained when we determined the relative contribution of ciliated and non-ciliated neurons in G2019S mutant dorsal striatum to overall *Gdnf* transcript production (*Figure 8D*). Unlike their wild-type counterparts, non-ciliated cholinergic neurons in LRRK2 G2019S striatal sections showed significantly higher levels of *Gdnf* gene expression, consistent with Hedgehog signaling dysregulation. The Hedgehog signaling pathway triggers *cilia-dependent* processing of the Gli3 protein: Gli3 can be converted into a transcriptional activator or it can be converted into a transcriptional repressor (*Bangs and Anderson, 2017*; *Kong et al., 2019*). It is possible that loss of Gli3 repression in non-ciliated LRRK2 mutant neurons increases GDNF production in those cells. Indeed, recent CHIP-Seq experiments identify the *Gdnf* gene as a Gli3-binding region, at least in the developing mouse limb bud (*Lex et al., 2020*). Finally, GDNF represses Sonic Hedgehog production by dopaminergic neurons (*Gonzalez-Reyes et al., 2012*); this may explain the decrease in *Gdnf* transcripts in ciliated LRRK2 mutant cholinergic neurons relative to wild-type animals, in response to the increased GDNF production by neighboring, non-ciliated LRRK2 mutant cholinergic neurons. Similar analysis of the relative contribution of ciliated cells to *Gli1* transcription in R1441C striatum (as in *Figure 6*) was consistent with a similar trend of dysregulation in unciliated mutant neurons (*Figure 8E*). Future experiments will seek to isolate these cells for single cell transcriptomic and/or proteomic analyses to determine the underlying basis for the differences observed for Hedgehog pathway components between ciliated and nonciliated mutant neurons.

## Phosphorylated Rab10 in astrocyte ciliogenesis

Our previous work in mouse embryonic fibroblasts and patient-derived iPS cells showed the importance of LRRK2-phosphorylated Rab10 and its effector, RILPL1 in primary ciliogenesis blockade (*Dhekne et al., 2018*). In addition, the fact that PPM1H mutant animals phenocopy hyperactive LRRK2 kinase mutants strongly implicates Rab GTPase phosphorylation in the phenotypes observed. Unfortunately, it was not possible to detect pRab10 directly in mouse brain sections. To explore further the possible contribution of pRab10 to ciliogenesis blockade in astrocytes, we used immunopanning methods (*Foo et al., 2011*) to obtain primary, poorly dividing astrocytes from G2019S⁺/⁻ LRRK2 rat brains. Cells are grown in defined, serum-free media containing 5 ng/ml soluble heparin binding EGF-like growth factor (HbEGF) that activates the EGF receptor (*Citri and Yarden, 2006*) and acts via the EGF receptor in these cells (*Foo et al., 2011*). Cell density is also relevant as cultured astrocytes secrete other autocrine trophic factors, and ciliogenesis is enhanced in cell culture by increased cell density.

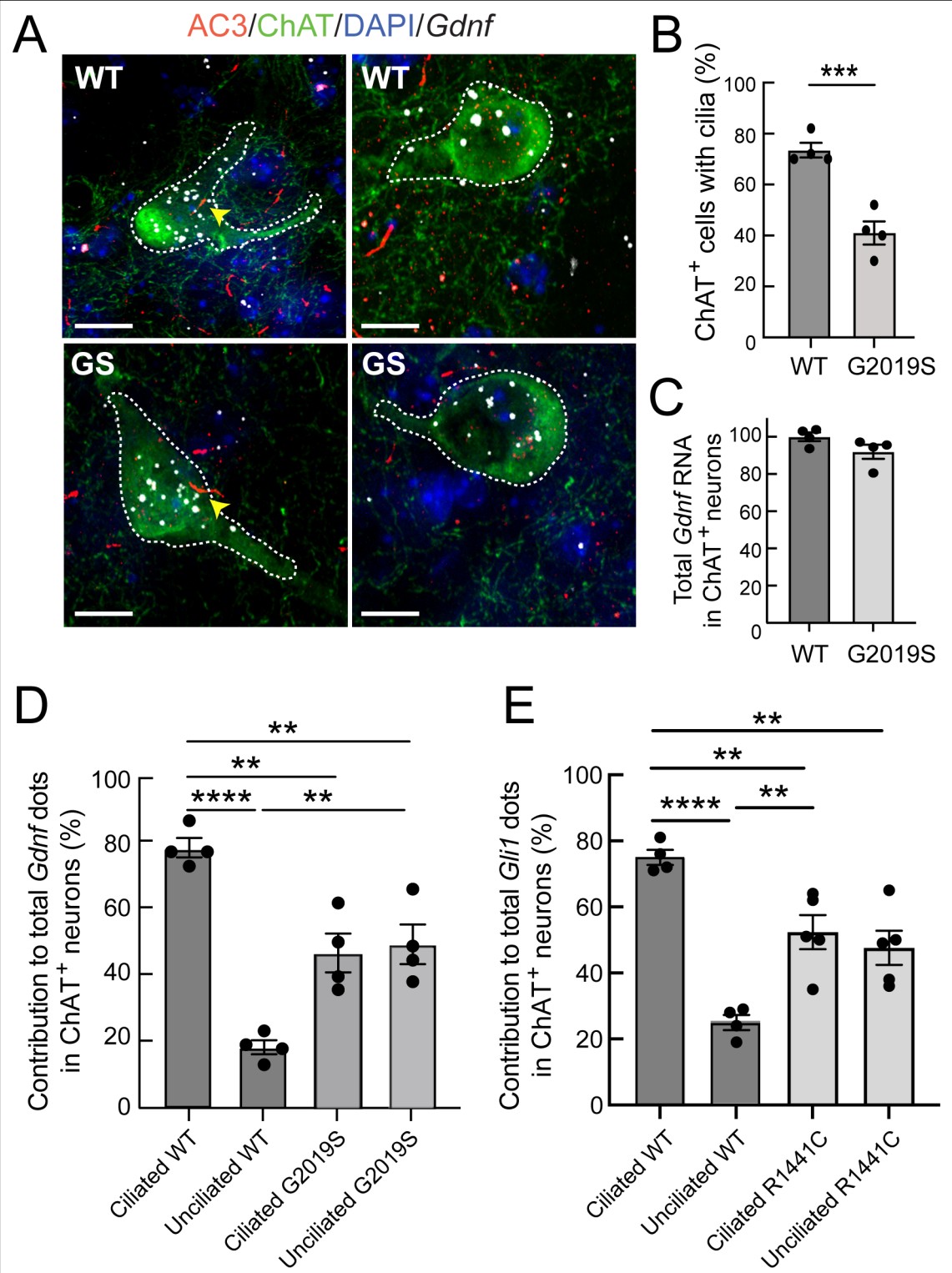

**Figure 8.** *Gdnf* expression is cilia dependent and dysregulated in LRRK2 G2019S striatum. (**A**) 5.5 month WT or LRRK2 G2019S mouse dorsal striatum was subjected to in situ hybridization using a *Gdnf* probe (white dots) and immunostained as indicated: ChAT (green, white outline), AC3 (red, yellow arrowheads), DAPI (blue). (**B**) Percent of cells with cilia. (**C**). Relative level of total *Gdnf* RNA in ChAT+ neurons as in A. (**D**) Relative contribution to total *Gdnf* dots over cholinergic interneurons as a function of their ciliation status. (**E**) Relative contribution to total *Gli1* dots over cholinergic interneurons as a function of their ciliation status in 10 month WT or R1441C mouse dorsal striatum (as in ***Figure 7***). Values represent the mean ± SEM from 4 WT and 4 G2019S brains each containing ≥33 cells. (**B**) WT vs G2019S, ***, p = 0.0009. (**D**) Ciliated WT vs Unciliated WT, ****, p < 0.0001. Ciliated WT vs Ciliated

*Figure 8 continued on next page*

Figure 8 continued

G2019S, **, p = 0.0020. Ciliated WT vs Unciliated G2019S; **, p = 0.0040. Unciliated WT vs Ciliated G2019S; **, p = 0.0040. Unciliated WT vs Unciliated G2019S; **, p = 0.0020. (**E**) Ciliated WT vs. Unciliated WT, ****, p < 0.0001. Unciliated WT vs Ciliated R1441C, **, p = 0.0029. Unciliated WT vs Unciliated R1441C, **, p = 0080; Ciliated WT vs Ciliated R1441C, **, p = 0.0080. Significance was determined either by student's t-test or by Ordinary one-way ANOVA using Dunnett's multiple comparisons test. Scale bars, 10 μm.

G2019S$^{+/-}$ panned astrocytes showed ~60 % ciliation when grown under sparse conditions (***Figure 9A and B***) as monitored using anti-Arl13B antibodies. Similar to what we have detected in multiple cell types in culture (***Steger et al., 2017***; ***Dhekne et al., 2018***), pRab10 levels were highest in non-ciliated cells in the absence of the MLi-2 LRRK2 inhibitor (***Figure 9B***), consistent with a correlation between pRab10 and ciliogenesis.

## Discussion

Pathogenic LRRK2 activity causes primary cilia loss in striatal cholinergic interneurons in mature R1441C LRRK2 mice (***Dhekne et al., 2018***). In this study, we show that primary cilia loss is also seen in two distinct G2019S LRRK2 mouse models of the most common human LRRK2 mutation. Moreover, our study reveals that primary cilia loss in cholinergic interneurons can be detected as early as

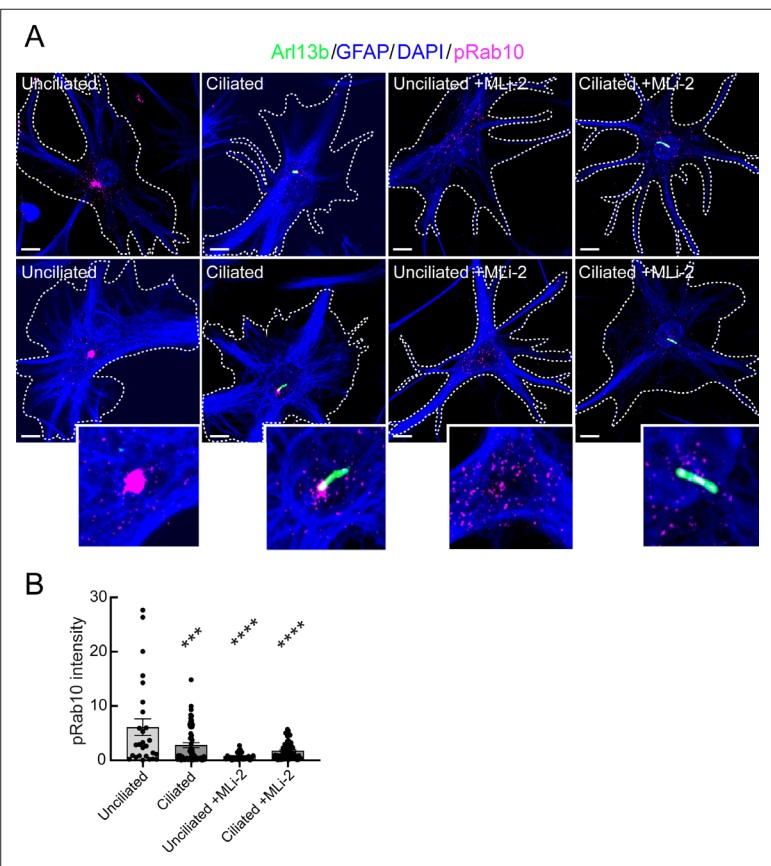

**Figure 9.** Immuno-panned primary G2019S LRRK2 astrocytes display increased pRab10. BAC Transgenic G2019S$^{+/-}$ LRRK2 rat astrocytes were dissected from P5 pups and cultured for 1 week ±200 nM MLi-2. (**A**). Astrocytes were labeled with anti-GFAP and DAPI (blue, white outline); rabbit-anti-phospho-Rab10 (magenta), and anti-Arl13B (green). The central portions of the bottom row images were enlarged and are shown at the lower right. Top row shows additional examples. (**B**) Quantitation of pRab10 intensity in ciliated or unciliated G2019S$^{+/-}$ astrocytes as in B. Unciliated G2019S$^{+/-}$ vs Ciliated G2019S$^{+/-}$; ***, p = 0.0009, Unciliated G2019S$^{+/-}$ vs MLi-2 treated Unciliated G2019S$^{+/-}$; ****, p < 0.0001; Unciliated G2019S$^{+/-}$ vs Ciliated MLi-2 treated; ****, p < 0.0001; ns = not statistically significant. Significance was determined either by student's t-test or by Ordinary one-way ANOVA using Dunnett's multiple comparisons test. Scale bars, 10 μm.

10 weeks of age, the youngest age we examined. Thus, pathogenic LRRK2 activity causes interneuron primary cilia loss at a relatively early stage and loss is sustained throughout the animal's lifespan for post-mitotic neurons. Consistent with this conclusion was our finding that mice deficient in the phospho-Rab-specific phosphatase, PPM1H, show the same ciliary defect as mice harboring a pathogenic LRRK2 mutation. This provides strong genetic evidence for the importance of LRRK2 activity in regulating cilia formation, in both LRRK2 wild-type/PPM1H-deficient and LRRK2 mutant animals.

Striatal cholinergic interneurons provide trophic support to dopaminergic neurons via Hh sensing (*Gonzalez-Reyes et al., 2012*), and primary cilia loss is predicted to perturb the ability of cholinergic neurons to sense Hh ligands as required for this role. Our FISH results support this model by showing that ciliated striatal ChAT[+] interneurons express *Gli1* transcripts under normal, physiologic conditions, and represent the cell type showing the greatest level of *Gli1* expression in the dorsal striatum. In the presence of the pathogenic R1441C LRRK2 mutation, we observed significant elevation of *Gli1* transcripts in the smaller number of remaining, ciliated, striatal cholinergic interneurons. One explanation for this increase in *Gli1* mRNA is that dopaminergic neurons in the substantia nigra upregulate Hh production due to stress or lack of GDNF signals from the striatum. Cortical neurons have been shown by others to require primary cilia to protect against stresses of ethanol- or ketamine-induced caspase activation and dendritic degeneration (*Ishii et al., 2021*). Future experiments will seek to monitor changes in Hh production in the substantia nigra that may be associated with LRRK2 mutation.

It was previously reported that loss of Hedgehog production by dopaminergic neurons decreases *Gdnf* transcripts and protein levels in the striatum in an age-dependent manner, with a 10-fold mRNA decrease seen between 1 and 12 months and a 35 % decrease in GDNF protein at 4 months of age (*Gonzalez-Reyes et al., 2012*). These authors assayed the entire striatum; it is important to note that significant GDNF is produced in the ventral striatum (*Barroso-Chinea et al., 2005*) and may account for some portion of the changes those authors observed. In our single-cell analyses of cells in the dorsal striatum, we did not detect a significant decrease in *Gdnf* transcripts at 5 months of age in LRRK2 G2019S striatal cholinergic neurons compared with age matched, wild-type controls. It will be important for us to analyze older mice as these showed a more significant change in the prior study, and to try to carry out GDNF protein analysis specifically in the dorsal striatum.

While transcription of the Hh target genes, *Gli1* and *Gdnf* were entirely cilia-dependent in wild-type cholinergic neurons, this selectivity was lost in G2019S and R1441C animals, where unciliated cells showed a much higher contribution to total *Gli1* and *Gdnf* transcript levels. Both of these genes are Gli3 repressor targets and one would have expected that the absence of a cilium would resemble a condition in which Gli3 repressor is high and target gene expression low. Future work will seek to clarify this unexpected dysregulation of the Hh signaling pathway.

L-dopa-induced dyskinesia is a debilitating side effect of dopamine replacement therapy for Parkinson's disease. *Malave et al., 2021* have shown that Smoothened activation in striatal cholinergic neurons protects mice from L-dopa-induced dyskinesia. These experiments highlight the importance of the Hedgehog signaling pathway for movement coordination regulated by the nigrostriatal circuit, and underscore the importance of studying the consequences of cilia loss associated with LRRK2 mutations since cilia are critical for Hedgehog signaling.

## Loss of cilia in astrocytes

For the first time, we show that LRRK2 G2019S striatal astrocytes are also cilia-deficient at 5.5–13 months of age and Gli levels correlate with the presence of primary cilia. The data indicate that fewer striatal astrocytes in LRRK2 G2019S mice are capable of responding to Hh ligands and importantly, the remaining astrocytes that do respond do not compensate for general loss in ciliated cells. How might decreased Hh sensing by astrocytes impact the striatum? Striatal astrocytes functionally interact with distinct networks of medium spiny neurons (*Khakh and Deneen, 2019*; *Martín et al., 2015*), and loss of Hh sensing may perturb these interactions. This is relevant because imbalances in medium spiny neuron activity are known to contribute to the slowness and rigidity of movement that is symptomatic of Parkinson's disease (*Zhai et al., 2018*).

*Chen et al., 2020* recently reported altered organization of glutaminergic AMPA receptors in cultured striatal neurons from LRRK2 G2019S and R1441C mutant mice. This was accompanied by decreased frequency of miniature excitatory post-synaptic currents in brain slices. It is likely that LRRK2-mediated Rab phosphorylation is in some way responsible for these changes; Rab phosphorylation

could of course regulate the trafficking of AMPA receptors. It is also possible that in the brain, overall physiological changes are due to loss of cilia from the astrocytes that surround these neurons and support their overall physiology.

Decreased Hh signaling in astrocytes results in the reduction of Kir4.1 potassium channel expression in the cerebellum and neocortex and impairs turnover of dendritic spines, accompanied by an increase in neuronal excitability (*Farmer et al., 2016*; *Hill et al., 2019*). Kir4.1 expression is markedly decreased in several neurological disorders (cf. *Scholl et al., 2009*; *Inyushin et al., 2010*; *Gilliam et al., 2014*), including in striatal cells of Huntington's disease mice (*Tong et al., 2014*; *Dvorzhak et al., 2016*). Striatal astrocytes may use Kir4.1 channels to modulate medium spiny neuron synaptic transmission by local buffering of potassium ions, as was shown for other brain regions (*Djukic et al., 2007*; *Sibille et al., 2014*). Thus, primary cilia loss from LRRK2 G2019S striatal astrocytes is likely to influence medium spinal neuron circuitry.

Two weeks of LRRK2 MLi-2 inhibitor administration failed to reverse cilia deficits in LRRK2 mutant animals. This may not be surprising as cilia are likely long-lived in post-mitotic neurons and non-reactive astrocytes. Very little is known about the dynamics of cilia in the adult brain; however, data from experiments in which a critical ciliary component, IFT88 was inducibly knocked out in adult brain indicated that 3 months was needed to see the consequences of loss of this ciliary factor (*Bowie and Goetz, 2020*). Thus, it is possible that longer feeding regimens will reveal a recovery in striatal ciliation. Another puzzle is why pRab10 levels do not change as significantly as pRab12 and LRRK2 pS935 in immunoblots of whole brains of drug-treated animals. It is possible that in the brain, pRab10 is greatly stabilized by strong effector binding. It is important to remember that levels of pRab10 in the very small population of cholinergic neurons of the dorsal striatum may be of major consequence for Hedgehog signaling in the nigrostriatal circuit and MLi-2 sensitivity of pRab10 in these cells would be missed in whole brain immunoblots.

Altogether, these data demonstrate a profound effect of LRRK2 hyperactivation on primary ciliogenesis in a population of cholinergic interneurons that play a key role in motor function and in astrocytes that support medium spiny neuron function in the dorsal striatum. Dysregulation of Hedgehog signaling by cholinergic interneurons and astrocytes of the striatum may contribute to the loss of dopaminergic neurons associated with a disease of aging such as Parkinson's disease.

## Materials and methods

**Key resources table**

| Reagent type (species) or resource | Designation | Source or reference | Identifiers | Additional information |
|---|---|---|---|---|
| Genetic reagent (*Rattus norvegicus*) | SA Sprague Dawley rat | Taconic | #SD-M | NTac:SD Background |
| Genetic reagent (*Rattus norvegicus*) | Human LRRK2 G2019S rat | Taconic | #10,681 | NTac:SD Background; BAC Transgene |
| Genetic reagent (*Mus musculus*) | Tg(LRRK2*G2019S)2AMjf | Jackson Laboratory | #018785 | C57BL/6 Background; BAC Transgene |
| Genetic reagent (*Mus musculus*) | Constitutive KI *Lrrk2*[tm4.1Arte] | Taconic | #13,940 | C57BL/6 Background; G2019S KI |
| Genetic reagent (*Mus musculus*) | B6.Cg-*Lrrk2*[tm1.1Shn]/J | Jackson Laboratory | #009346, RRID:IMSR_JAX:009346 | C57BL/6 Background; R1441C KI |
| Genetic reagent (*Mus musculus*) | *Ppm1h*[-/-] mouse | Taconic | #TF3142 | C57BL/6 Background |
| Antibody | anti-Arl13B (mouse monoclonal) | Neuromab | N295B/66 | (1:1000) |
| Antibody | anti-Adenylate cyclase III (rabbit polyclonal) | Santa Cruz | SC-588 | (1:100) |
| Antibody | anti-Adenylate cyclase III (rabbit polyclonal) | EnCOR | RPCA-ACIII | (1:10000) |

*Continued on next page*

*Continued*

| Reagent type (species) or resource | Designation | Source or reference | Identifiers | Additional information |
|---|---|---|---|---|
| Antibody | anti-NeuN (mouse monoclonal) | Proteintech | 66836–1-IG | (1:500) |
| Antibody | anti-Choline Acetyltransferase (goat polyclonal) | Millipore | AB144P-1ML | (1:100) |
| Antibody | anti-GFAP (chicken polyclonal) | Novus Biologicals | NBP1-05198 | (1:2000) |
| Antibody | anti-GFAP (chicken polyclonal) | EnCOR | CPCA-GFAP | (1:2000) |
| Antibody | anti-S100B (guinea pig polyclonal) | Synaptic Systems | #287,004 | (1:200) |
| Antibody | anti-phospho-Rab10 (Thr73) (rabbit monoclonal) | Abcam | AB230261 | (1:1000) |
| Antibody | anti-Rab10 (mouse monoclonal) | Nanotools | 0680–100/Rab10-605B11 | (1:1000) |
| Antibody | anti-phospho-Rab12 (Ser105) (rabbit monoclonal) | Abcam | ab256487 | (1:1000) |
| Antibody | anti-Rab12 (sheep polyclonal) | MRC PPU Reagents and Services, University of Dundee | SA227 | (1:1000) |
| Antibody | Anti-phospho-LRRK2 (Ser935) (rabbit monoclonal) | MRC PPU Reagents and Services, University of Dundee | UDD2 | (1:1000) |
| Antibody | Anti-LRRK2 (mouse monoclonal) | Antibodies Inc./NeuroMab | 75–253 | (1:1000) |
| Antibody | anti-PPM1H (sheep polyclonal) | MRC PPU Reagents and Services, University of Dundee | DA018 | (1:1000) |
| Chemical compound, drug | MLi-2 | MRC PPU Reagents and Services, University of Dundee | | |
| Commercial assay or kit | RNAscope Multiplex Fluorescent Reagent Kit v2 | Advanced Cell Diagnostics | #323,100 | |
| Commercial assay or kit | RNAscope 3-plex Negative Control Probe | Advanced Cell Diagnostics | #320,871 | |
| Commercial assay or kit | RNAscope Probe - Mm-*Gli1* | Advanced Cell Diagnostics | #311,001 | |
| Commercial assay or kit | RNAscope Probe - Mm-*GDNF* | Advanced Cell Diagnostics | #421,951 | 1:10 |
| Commercial assay or kit | OPAL 690 REAGENT PACK | Akoya Biosciences | FP1497001KT | (1:750) |
| Other | | Research Diets, Inc. | D01060501 | Untreated diet |
| Other | | Research Diets, Inc. | D01060501 added with 360 mg MLi-2 per kg | Modified diet |
| Software, Algorithm | FIJI | PMID:29187165 | RRID:SCR_002285 | |
| Software, Algorithm | CellProfiler | PMID:29969450 | RRID:SCR_007358 | |

## Key resources
Primary antibodies and reagents used in this study.

## Reagents
MLi-2 LRRK2 inhibitor was synthesized by Natalia Shpiro (MRC Reagents and Services, University of Dundee) and was first described to be a selective LRRK2 inhibitor in previous work (*Fell et al., 2015*).

## Transgenic mice

All animal studies were performed in accordance with the Animals (Scientific Procedures) Act of 1986 and regulations set by the University of Dundee, the U.K. Home Office, and the Administrative Panel on Laboratory Animal Care at Stanford University. Mice were maintained under specific pathogen-free conditions at the University of Dundee (U.K.); they were multiply housed at an ambient temperature (20–24°C) and humidity (45–55%) and maintained on a 12 hr light/12 hr dark cycle, with rodent diet and water available ad libitum. LRRK2 R1441C knock-in mice backcrossed on a C57BL/6 J background, were obtained from the Jackson laboratory (Stock number: 009346). LRRK2 G2019S knock-in and PPM1H knock-out mice backcrossed on a C57BL/6 J background, were obtained from Taconic (Model 13,940 and TF3142, respectively). G2019S BAC transgenic brain sections from 10 month mice were a gift from Aaron Gitler at Stanford (*Li et al., 2010*). Mouse genotyping was performed by PCR using genomic DNA isolated from tail clips or ear biopsies. For the experiment shown in *Figure 5*, *Figure 5—figure supplement 1*, 63–83 day old mice of the indicated genotypes were injected subcutaneously with vehicle [40 % (w/v) (2-hydroxypropyl)-β-cyclodextrin (Sigma–Aldrich #332607)] or MLi-2 dissolved in the vehicle at a 30 mg/kg final dose. Mice were euthanized by cervical dislocation 2 hr following treatment and the collected tissues were rapidly snap frozen in liquid nitrogen.

## Mouse brain processing

Homozygous LRRK2-mutant (R1441C or G2019S of ages indicated) or heterozygous $Ppm1h^{-/+}$ mice (164 days old) and age-matched wild type controls were fixed by transcardial perfusion using 4 % paraformaldehyde (PFA) in PBS as described in *Khan et al., 2020*. Next, whole brain tissue was extracted, post-fixed in 4 % PFA for 24 hr and then immersed in 30 % (w/v) sucrose in PBS until the tissue settled to the bottom of the tube (~48 hr). LRRK2 R1441C KI, LRRK2 G2019S KI and $Ppm1h^{-/+}$ brains were harvested in Dundee and sent with identities blinded until analysis was completed. Prior to cryosectioning, brains were embedded in cubed-shaped plastic blocks with OCT (BioTek, USA) and stored at −80 °C. OCT blocks were allowed to reach −20 °C for ease of sectioning. The brains were oriented to cut sagittal or coronal sections on a cryotome (Leica CM3050S, Germany) at 16–25 μm thickness and positioned onto SuperFrost plus tissue slides (Thermo Fisher, USA).

## In-diet MLi-2 administration to wild-type mice (Pilot study) and LRRK2 R1441C KI mice

For the experiment shown in *Figure 4*, 11 C57BL/6 J wild-type mice were allowed to acclimate to the control rodent diet (Research Diets D01060501; Research Diets, New Brunswick, NJ) for 14 days before being placed on study. On day 1 of the study, one group (4 mice) received modified rodent diet (Research Diets D01060501) containing MLi-2 and formulated by Research Diets to provide a concentration of 60 mg/kg per day on the basis of an average food intake of 5 g/day for 14 days; the other group (7 mice) received untreated diet (Research Diets D01060501) for 14 days and served as the control group. The dose of MLi-2 and the length of the in-diet treatment used for this study were based on *Fell et al., 2015*. Bodyweight and food intake were assessed twice weekly. On the last day of the study, 3 mice from the control diet group received 30 mg/kg MLi-2 dissolved in 40 % (w/v) (2-hydroxypropyl)- β-cyclodextrin via subcutaneous injection for 2 hr prior to tissue collection. All mice were euthanized by cervical dislocation and the collected tissues were rapidly snap frozen in liquid nitrogen and brain samples used for quantitative immunoblotting analysis of phospho-Ser935 LRRK2 and phospho-Ser105 Rab12 as a readout of LRRK2 activity.

For the experiments described in *Figure 3C–G*, littermate or age-matched male and female LRRK2 R1441C homozygous knock-in mice at 10 weeks of age were used. Mice were allowed to acclimate to the control rodent diet for 14 days as described above before being placed on study. On day 1 of the study, one group (7 mice) received a modified rodent diet targeted to provide a concentration of 60 mg/kg per day on the basis of an average food intake of 5 g/day; the other group (7 mice) received an untreated diet and served as the control group. Bodyweight and food intake were assessed twice weekly. On day 15, mice were terminally anesthetized and brains harvested and fixed as described above. For immunoblotting analysis to confirm LRRK2 inhibition, brains were snap frozen in liquid nitrogen from mice that were terminally anesthetized and perfused by injection of PBS into the left cardiac ventricle (2 mice from each group).

## Preparation of mouse tissue lysates for immunoblotting analysis

Snap frozen tissues were weighed and quickly thawed on ice in a 10-fold volume excess of ice-cold lysis buffer containing 50 mM Tris–HCl pH 7.4, 1 mM EGTA, 10 mM 2-glycerophosphate, 50 mM sodium fluoride, 5 mM sodium pyrophosphate, 270 mM sucrose, supplemented with 1 µg/ml micro-cystin-LR, 1 mM sodium orthovanadate, complete EDTA-free protease inhibitor cocktail (Roche), and 1 % (v/v) Triton X-100. Tissues were homogenized using a POLYTRON homogenizer (KINEMATICA), employing three rounds of 10 s homogenization with 10 s intervals on ice. Lysates were clarified by centrifugation at 20,800 g for 30 min at 4 °C and supernatant was collected for subsequent protein quantification by Bradford assay and immunoblot analysis.

## Quantitative immunoblotting analysis

40 µg of brain extracts were loaded onto a NuPAGE 4–12% Bis–Tris Midi Gel (Thermo Fisher Scientific, Cat# WG1402BOX) and electrophoresed at 130 V for 2 hr with NuPAGE MOPS SDS running buffer (Thermo Fisher Scientific, Cat# NP0001-02). Proteins were then electrophoretically transferred onto a nitrocellulose membrane (GE Healthcare, Amersham Protran Supported 0.45 µm NC) at 100 V for 90 min on ice in transfer buffer (48 mM Tris–HCl and 39 mM glycine supplemented with 20 % (v/v) methanol). The transferred membrane was blocked with 5 % (w/v) skim milk powder dissolved in TBS-T (50 mM Tris base, 150 mM sodium chloride (NaCl), 0.1 % (v/v) Tween 20) at room temperature (RT) for 30 min and incubated overnight at 4 °C in primary antibodies diluted in 5 % (w/v) BSA in TBS-T. After incubation with primary antibodies, membranes were washed three times for 5 min with TBS-T and incubated with near-infrared fluorescent dye-labelled secondary antibodies (diluted to 1:20,000) for 1 hr at RT. Thereafter, membranes were extensively washed with TBS-T and protein bands were acquired via near-infrared fluorescent detection using the Odyssey CLx imaging system and the signal intensity quantified using Image Studio software.

## Fluorescence in situ hybridization (FISH)

Wild-type (WT) and R1441C LRRK2 KI 10 month mouse brains were sliced coronally at 25 µm thick-ness. RNAscope Multiplex Fluorescent Detection Kit v2 (Advanced Cell Diagnostics) was carried out according to the manufacturer using RNAscope 3-plex Negative Control Probe (#320871) or Mm-*Gli1* (#311001). Opal 690 (Akoya Biosciences) was used for fluorescent visualization of hybrid-ized probes. Then, brain slices were blocked with 0.1 % BSA and 10 % donkey serum in TBS (Tris buffered saline) containing 0.1 % Triton X-100 for 30 min followed by incubation with primary anti-body in TBS +0.1 % BSA and 1 % DMSO overnight at 4 °C. Secondary antibody was also diluted in TBS +0.1 % BSA and 1 % Triton X-100 for 30 min and then added for 2 hr at RT. Sections were mounted with ProLong Gold Antifade Mountant with DAPI and glass coverslips. For *Gdnf* RNA experiments, WT and G2019S LRRK2 KI 5 month mouse sections were sliced coronally at 25 µm thickness. RNAscope Multiplex Fluorescent Detection was performed as above using Mm *Gdnf* (#421951) diluted 1:10 in dilution buffer (6 x saline-sodium citrate buffer (SSC), 0.2 % lithium dodecylsulfate (Research Products International; #RPI-L26200), and 20 % Calbiochem OmniPur Formamide (#75-12-7)).

## Immunofluorescence staining and microscopy

Brain primary cilia detection and analyses were performed as previously described (*Khan et al., 2020*). Briefly, frozen slides were thawed at RT for 10 min then gently washed with PBS for 5 min. Sections were permeabilized with 0.1 % Triton X-100 in PBS at RT for 15 min. Sections were blocked with 2 % BSA in PBS for 2 hr at RT and were then incubated overnight at 4 °C with primary antibodies. The following day, sections were incubated with secondary antibodies at RT for 2 hr. Donkey highly cross-absorbed H + L secondary antibodies (Life Technologies) conjugated to Alexa 488, Alexa 568 or Alexa 647 were used at a 1:1000 dilution. Stained tissues were overlayed with Mowiol and a glass coverslip. All antibody dilutions for tissue staining included 1 % DMSO to help antibody penetration. All images were obtained using a spinning disk confocal microscope (Yokogawa) with an electron multiplying charge coupled device (EMCCD) camera (Andor, UK) and a 100 × 1.4 NA oil immersion objective. All image visualizations and analyses were performed using Fiji (RRID:SCR_002285; *Rueden et al., 2017*). Cholinergic neurons in the dorsal striatal region were identified in the caudate putamen of sagittal or coronal sections that were reactive for both choline acetyltransferase and the pan neuronal marker, NeuN. Dorsal striatal astrocytes were defined as GFAP[+] and/or S100B[+] cells in the caudate putamen.

## Astrocyte immune-panning and microscopy

Rat Astrocyte Immune-panning was performed as previously described (*Foo et al., 2011*; *Dhekne et al., 2018*). Cells on coverslips were fixed with 3.5 % PFA for 15 min at RT. The cells were then subjected to three washes with PBS for 5 min each. To permeabilize, samples were incubated with PBS containing 0.1 % Saponin for 15 min. All subsequent steps contained 0.05 % Saponin unless otherwise specified. Samples were again washed three times with PBS then incubated in blocking solution (PBS containing 2 % BSA) for 1 hr at RT. Afterwards, the cells were incubated in blocking solution containing primary antibodies for 1 hr at RT (EnCOR Chicken-anti-GFAP at 1:2000, Abcam Rabbit-anti-pRab10 at 1:1000, Neuromab Mouse-anti-Arl13B at 1:2000). Cells were then washed three times with blocking solution, then incubated for 1 hr at RT with blocking solution containing DAPI and donkey-anti alexa fluor secondary antibodies. Coverslips were then rinsed without saponin in PBS two times, in ddH$_2$O once, and then mounted with mowiol. All image visualizations and analyses were performed using Fiji (RRID:SCR_002285). Maximum intensity projections, background subtraction, and pRab10 integrated intensity measurements were obtained using CellProfiler (RRID:SCR_007358; *McQuin et al., 2018*).

## Statistics

Graphs were made using Graphpad Prism six software (GraphPad Prism, RRID:SCR_002798). Error bars indicate SEM. Unless otherwise specified, a Student's T-test was used to test significance. Brains were harvested in Dundee and analyzed at Stanford; identities were blinded for unbiased analyses. RNA hybridization was quantified using CellProfiler (RRID:SCR_007358; *McQuin et al., 2018*).

## Acknowledgements

This study was funded by the joint efforts of The Michael J Fox Foundation for Parkinson's Research (MJFF) [17298 & 6,986 (SR P & DRA)] and Aligning Science Across Parkinson's (ASAP) initiative. MJFF administers the grant (ASAP-000463, SRP & DRA) on behalf of ASAP and itself. Funds were also provided by the Medical Research Council [grant no. MC_UU_12016/2 (DRA)] and the pharmaceutical companies supporting the Division of Signal Transduction Therapy Unit (Boehringer-Ingelheim, GlaxoSmithKline, Merck KGaA (DRA)). For the purpose of open access, the authors have applied a CC-BY public copyright license to the Author Accepted Manuscript version arising from this submission. All primary data associated with each figure has been deposited in the Dryad repository and can be found at https://doi.org/10.5061/dryad76hdr7sxx. We are especially grateful to Drs. Rajat Rohatgi and Jennifer Kong for their critical input and Dr. Andreas Kottman for helpful discussion.

## Additional information

### Funding

| Funder | Grant reference number | Author |
| --- | --- | --- |
| Michael J. Fox Foundation for Parkinson's Research | 17298 & 6986 | Shahzad S Khan<br>Yuriko Sobu<br>Herschel S Dhekne<br>Francesca Tonelli<br>Kerryn Berndsen<br>Dario R Alessi<br>Suzanne R Pfeffer |
| Aligning Science Across Parkinson's | ASAP-000463 | Shahzad S Khan<br>Yuriko Sobu<br>Herschel S Dhekne<br>Francesca Tonelli<br>Kerryn Berndsen<br>Dario R Alessi<br>Suzanne R Pfeffer |
| National Institutes of Health | DK37332 | Suzanne R Pfeffer |
| Medical Research Council | MC_UU_12016/2 | Dario R Alessi |

| Funder | Grant reference number | Author |
|---|---|---|

The funders had no role in study design, data collection and interpretation, or the decision to submit the work for publication.

## Author contributions

Shahzad S Khan, Conceptualization, Data curation, Formal analysis, Investigation, Validation, Visualization, Writing - original draft, Writing - review and editing; Yuriko Sobu, Conceptualization, Investigation, Methodology, Validation, Visualization, Writing - review and editing; Herschel S Dhekne, Conceptualization, Data curation, Investigation, Writing - review and editing; Francesca Tonelli, Investigation, Project administration, Resources, Writing - review and editing; Kerryn Berndsen, Investigation, Validation; Dario R Alessi, Conceptualization, Data curation, Funding acquisition, Project administration, Supervision, Writing - review and editing; Suzanne R Pfeffer, Conceptualization, Data curation, Formal analysis, Funding acquisition, Investigation, Project administration, Supervision, Visualization, Writing - original draft, Writing - review and editing

## Author ORCIDs

Shahzad S Khan ![orcid] http://orcid.org/0000-0003-3962-0226
Herschel S Dhekne ![orcid] http://orcid.org/0000-0002-2240-1230
Francesca Tonelli ![orcid] http://orcid.org/0000-0002-4600-6630
Kerryn Berndsen ![orcid] http://orcid.org/0000-0002-9353-7565
Dario R Alessi ![orcid] http://orcid.org/0000-0002-2140-9185
Suzanne R Pfeffer ![orcid] http://orcid.org/0000-0002-6462-984X

## Ethics

All animal studies were performed in accordance with the Animals (Scientific Procedures) Act of 1986 and regulations set by the University of Dundee, the U.K. Home Office, and the Administrative Panel on Laboratory Animal Care at Stanford University.

## Decision letter and Author response

Decision letter https://doi.org/10.7554/eLife.67900.sa1
Author response https://doi.org/10.7554/eLife.67900.sa2

# Additional files

## Supplementary files
• Transparent reporting form

## Data availability

All primary data associated with each figure has been deposited in the Dryad repository and can be found at https://doi.org/10.5061/dryad.76hdr7sxx.

The following dataset was generated:

| Author(s) | Year | Dataset title | Dataset URL | Database and Identifier |
|---|---|---|---|---|
| Pfeffer SR | 2021 | Data from: Pathogenic LRRK2 control of primary cilia and Hedgehog signaling in neurons and astrocytes of mouse brain | https://doi.org/10.5061/dryad.76hdr7sxx | Dryad Digital Repository, 10.5061/dryad.76hdr7sxx |

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
