## [Decision Letter]

**Acceptance summary:**

This is an interesting follow-up to a paper recently published in e*Life* reporting the discovery of a phosphatase (PPM1H) that can remove phosphate from sites of modification in Rabs that are phosphorylated by LRRK2. This new work demonstrates that LRRK2 mutants are incapable of building cilia in cholinergic interneurons of the dorsal striatum. Interestingly, this phenotype is recapitulated in the PPM1H mouse mutant, clearly implicating LRRK2-dependent phosphorylation of specific Rabs as critical for the formation/maintenance of cilia. As expected, this cilia phenotype correlates with a defect in hedgehog signaling in vivo. Because LRRK2 mutants have increased levels of Rab phosphorylation, as do PPM1H mutants, this data fits nicely with the idea that the status of Rab phosphorylation is having input on cilia.

**Decision letter after peer review:**

Thank you for resubmitting your work entitled "Pathogenic LRRK2 control of primary cilia and Hedgehog signaling in neurons and astrocytes of mouse brain" for further consideration by *eLife*. Your article has been reviewed by 3 peer reviewers, one of whom is a member of our Board of Reviewing Editors, and the evaluation has been overseen by a Reviewing Editor. The following individual involved in review of your submission has agreed to reveal their identity: Peter McPherson (Reviewer #2)

The manuscript has many strengths and the reviewers are generally quite positive but there are some remaining issues that need to be addressed, as outlined below:

1. It would make the case a bit stronger if the authors could look at the transcript levels of GDNF from the same RNA samples they used to analyze Gli1.

2. The work is of immense importance and experiments were well performed. However, more clarification in the text would be required if not more experiments, on the possible effect(s) of elevated levels of Gli1 transcript in the striatum. Additional work would allow the authors to get a sense of whether/how the changes in Gli levels that they report for the ciliated vs non ciliated translate to changes at the population level for each of the cell types. The authors suggest how they think it works, but it would be nice to actually show it. For example do the cholinergic neurons overall have reduced Hh output despite the fact that the cells that remain ciliated increase their response? This seems important in terms of understanding what is happening at the tissue level.

3. Clarify the timing of the LRRK2 inhibitor experiments and why there was, and especially why there is not much of a change in Rab10 phosphorylation. Try to strengthen the link between Rab10 phosphorylation and reduced cilia.

In order to facilitate the revision, we have included all of the comments from the reviewers for a perspective but you should focus on the list above. We look forward to seeing your revised paper.

*Reviewer #1:*

This is an interesting follow-up to a paper recently published in *eLife* reporting the discovery of a phosphatase (PPM1H) that can remove phosphate from sites of modification in Rabs that are phosphorylated by LRRK2. This new work demonstrates that LRRK2 mutants are incapable of building cilia in cholinergic interneurons of the dorsal striatum. Interestingly, this phenotype is recapitulated in the PPM1H mouse mutant, clearly implicating LRRK2-dependent phosphorylation of specific Rabs as critical for the formation/maintenance of cilia. As expected, this cilia phenotype correlates with a defect in hedgehog signaling in vivo. Because LRRK2 mutants have increased levels of Rab phosphorylation, as do PPM1H mutants, this data fits nicely with the idea that the status of Rab phosphorylation is having input on cilia.

The primary phenotypes observed reflect a reduction (generally ~50%) in cilia number for the various LRRK2 and PPM1H mutants examined in both interneurons and astrocytes. In cells that maintained cilia, the length of the cilia were largely unchanged. This raises the question about what is different among the cells that did and did not generate a cilia. But addressing this question is likely beyond the scope of this paper.

Overall, the data as far as it goes is well presented and controlled. Replicate experiments are included at every stage, indicating the robustness of the results.

So in my view, this paper fits well into the Research Advances category in that it provides a strong indication that PPM1H functions biologically in the same pathway as LRRK2 mutation.

*Reviewer #2:*

This work builds on Dhekne et al., 2018, demonstrating decreased ciliation of cholinergic neurons in the striatum of LRRK2 R1441C mice. The findings cause the authors to speculate that such decreased ciliation phenotype might lead to a decrease in sonic hedgehog (Hh) response, which might further impact a neuro-protective circuit that supports dopaminergic neurons.

In this study, the authors provide evidence to support the idea of reduced Hh response and report the following important findings:

1) G2019S LRRK2 mice also show loss of cilia in striatal cholinergic interneurons. This finding was also supported by the data showing loss of primary cilia in the phospho-Rab-specific phosphatase, PPM1H deficient mice. As PPM1H specifically dephosphorylates LRRK2 substrates, collectively it strengthens the fact that the phenotype is caused by LRRK2 mutations (both R1441C and G2019S).

2) Although the number of cholinergic cells showing cilia are less, Gli1 transcripts were found to be elevated in those smaller number of cells. A similar pattern was noted in astrocytes. The authors suggest that the overall loss of ciliation due to the LRRK2 mutation will lead to an overall decrease in striatal astrocytic Hh response.

3) In addition, striatal astrocytes were cilia-deficient, leading to new hypotheses to be tested in future in terms of the current knowledge of how astrocytes support neurons with respect to Parkinson's model.

While this study tests an important idea of how defect in ciliogenesis could lead to dopaminergic neuron loss based on the current knowledge of circuits between dopaminergic neurons in the substantia nigra and cholinergic interneurons in the striatum (Gonzalez-Reyes et al., (2012)), it has a missing component that should be addressed in order to establish this model:

It would be important to measure GDNF expression in the striatum. As the authors speculate that there is an overall decrease in Hh response due to less cilia, this statement will carry more weight if the authors show that there is an inhibition of GDNF expression. Ideally this would be done by measuring secreted GDNF. However, this may be technically very demanding. Alternatively, the authors could examine GDNF transcripts. Altered GDNF transcript/expression/release is more likely to cause damage to the dopaminergic neurons. As there is an elevated level of Gli1 transcripts in a smaller number of ciliated cells (cholinergic) in R1441C mice, it raises concerns about how much of DA loss might happen when there is an elevated level of Gli1. Addressing the question on GDNF could solve the issue.

It would really build the study if you could examine GDNF expression, as is done in figure 6B of the Gonzalez-Reyes et al., (2012) manuscript. I do understand if this is too technically demanding. Alternatively, one could examine GDNF transcripts.

Are there humans with PPM1H variants? NCBI's Clinvar database is a very good site to explore this.

"Rab12 S105 phosphorylation also decreased in wild type (Figure 4C) and mutant (Figure 4A, B) brain (Figure 4). However, levels of pRab10 seemed unchanged (Figure 4A, B)". I don't really understand this seeming discrepancy. I just wonder if there is anything new that could be added.

Technical problems:

1) Line 65, Dhekne et al., 2021 should be 2018?

2) Figure 6 D, statistical significance is required. Or, the authors mean that there is an increasing trend but not statistically significant.

3) Figure 8 C, representative images would help if possible.

*Reviewer #3:*

Khan et al., is a Research Advance to a publication from the group in 2018. The work provides confirmation that a more common variant associated with PD in humans also results in a similar reduction in cilia frequency in cholinergic interneurons in the striatum. This was noted in one figure of the 2018 paper for a mouse knockin of the R1441C variant of LRRK2. In the current work they use both a knockin and a BAC overexpression of another LRRK2 variant, G2019S, to assess further the impact of the LRRK2 mutations on cilia in the striatum. They examine the frequency of cilia in ChAT+ striatal interneurons and astrocytes in mice with two different PD-associated mutations, and they also investigate SHH signaling in these cells. While the paper is an interesting advance over the prior work, I think some additional experiments and clarifications are needed to improve the work prior to publication.

A strength of the paper is that it further builds upon the prior work of this group on the mechanisms by which LRRK2 activating mutations suppress cilia by extending their findings to specific neurons within the brain. They show that cilia and ciliary signaling in cholinergic interneurons and astrocytes are affected by activating mutations in LRRK2. This is an important step in building the connection between the activation of the kinase, ciliary defects, and the actual neurodegenerative outcomes in patients.

However, a weakness of the paper is that the conclusions with regard to both phosphorylation of Rab10 in this in vivo context and SHH signaling extend a bit beyond the data and these gaps should be addressed.

Major points:

The LRRK2 inhibitor experiment in the PPM1H^+^/- mice in Figure 5 is confusing- why is this done such a short time beforehand here when in other experiments in Figure 4 the animals were fed the inhibitor for 2 weeks. It's also confusing that the text says "they strongly validate a link between LRRK2-Rab phosphorylation and ciliogenesis in specific brain cell types", however the inhibitor experiment doesn't show much change in RAB10 phosphorylation in the PPM1H mice +/- the inhibitor or even the PPM1H mice relative to WT. Some quantification here is necessary to better investigate these changes. And some work could also be done to better clarify the point the authors are trying to make here. More generally, the connection between Rab10 phosphorylation in vivo and the reduction in cilia frequency seems unclear. Is phosphorylation of RAB10 increased in the brain of the R1441C mice compared with WT? Only the knockin mice are shown, and I don't see this in the previous paper.

The Hh signaling data also needs some development and clarification. For the cholinergic neurons- are the overall levels of Gli1 affected? Could this be measured by RNAseq?

Another way to look at Hh signaling in the cholinergic neurons would be to look at Hh pathway components in cilia. SMO antibodies in particular often work well even in tissue. The authors could test whether SMO localization or intensity in the cilia of the cholinergic neurons is affected which would more directly address whether the cilia that are present are altered in their function as the authors posit.

The authors also state that reduced numbers of cilia in astrocytes of the R1441C mice leads to an overall reduction in Hh signaling in these cells. I understand they are inferring this from the fact that there are fewer ciliated astrocytes mice but it should also be shown directly.

---

## [Author Response]

The manuscript has many strengths and the reviewers are generally quite positive but there are some remaining issues that need to be addressed, as outlined below:1. It would make the case a bit stronger if the authors could look at the transcript levels of GDNF from the same RNA samples they used to analyze Gli1.

We thank the reviewers for this suggestion. As requested, we have now used RNAscope fluorescence in situ hybridization to measure GDNF transcripts in cholinergic neurons from dorsal striatal tissue of 5-month old wild type and G2019S LRRK2 mutant mice (New Figure 8). Our results show that GDNF transcripts are restricted primarily to cholinergic neurons of the dorsal striatum. Moreover, we find that in wild type animals, most GDNF transcripts are detected in ciliated cells. At first glance this would indicate that cilia-deficient LRRK2 mutant neurons would produce fewer GDNF transcripts. Surprisingly, however, in G2019S mutant mice, we detect a much greater proportion of GDNF transcripts in *unciliated* striatal cholinergic neurons compared with wild type neurons consistent with Hedgehog signaling dysregulation. We fail to detect a decrease in total GDNF transcripts, but Hedgehog dysregulation is very clear and seen in the cholinergic neurons and not in adjacent astrocytes.

The GDNF gene is a target of the Gli3 repressor (at least in the limb bud). Our data indicate that in the absence of cilia, cells lose *cilia-dependent* Gli3-repression of GDNF expression in G2019S striatal CIN. In previous work, Kottmann and colleagues (Gonzalez-Reyes et al., (2012)) noted a strong age dependence of GDNF loss, with the biggest effect at 12 months. Also, in that study, both dorsal and ventral striatal tissue was analyzed--here we have carried out single cell analyses of dorsal striatum only. Future experiments will determine whether older LRRK2 mutant mice show a further decrease in GDNF expression.

2. The work is of immense importance and experiments were well performed. However, more clarification in the text would be required if not more experiments, on the possible effect(s) of elevated levels of Gli1 transcript in the striatum. Additional work would allow the authors to get a sense of whether/how the changes in Gli levels that they report for the ciliated vs non ciliated translate to changes at the population level for each of the cell types. The authors suggest how they think it works, but it would be nice to actually show it. For example do the cholinergic neurons overall have reduced Hh output despite the fact that the cells that remain ciliated increase their response? This seems important in terms of understanding what is happening at the tissue level.

The reviewers ask us to clarify the consequences of changes in ciliary Hh signaling, and we agree this is important. In our new analyses, we quantify the contribution of ciliated and non ciliated cells to both total Gli1 and GDNF transcript production. Re-analysis of our data indicates that similar to what we see for GDNF, Gli1 response is strongly cilia dependent in wild type neurons but less cilia-dependent overall in mutant neurons. This is likely due to the same loss of cilia-dependent Gli3 repression seen for GDNF (point #1 above).

Interestingly we see a neuron-specific dysregulation that is not seen in adjacent astrocytes that retain cilia-dependent Gli1 responses.

3. Clarify the timing of the LRRK2 inhibitor experiments and why there was, and especially why there is not much of a change in Rab10 phosphorylation. Try to strengthen the link between Rab10 phosphorylation and reduced cilia.

Part of the reason it is hard to see decreases in Rab10 phosphorylation in brain is that there is very little unless one looks in PPM1H homozygous knock out brains (now shown in this revision). Four other reports have seen the same effect of apparent insensitivity of brain pRab10 to MLi treatment. The whole brain was analyzed by western blot and remember that the cholinergic interneurons that we study in the striatum represent only ~5% of the total cells present in that region; pRab10 changes or sensitivity to MLi-2 in that region cannot yet be monitored. Thus, current methods make it impossible to provide a direct connection between Rab phosphorylation and ciliation in cholinergic interneurons of the dorsal striatum. HOWEVER, this is why analysis of Rab-specific PPM1H phosphatase knockouts is so important--these animals phenocopy activating LRRK2 mutations, strongly supporting the idea that Rab phosphorylation is playing a critical role. In primary cultured astrocytes we see a clear correlation and this is also true for every other cell type analyzed. Future work will try to carry out mass spec determinations of pRab10 in small numbers of isolated, cholinergic striatal neurons.

In order to facilitate the revision, we have included all of the comments from the reviewers for a perspective but you should focus on the list above. We look forward to seeing your revised paper.Reviewer #2:While this study tests an important idea of how defect in ciliogenesis could lead to dopaminergic neuron loss based on the current knowledge of circuits between dopaminergic neurons in the substantia nigra and cholinergic interneurons in the striatum (Gonzalez-Reyes et al., (2012)), it has a missing component that should be addressed in order to establish this model:It would be important to measure GDNF expression in the striatum. As the authors speculate that there is an overall decrease in Hh response due to less cilia, this statement will carry more weight if the authors show that there is an inhibition of GDNF expression. Ideally this would be done by measuring secreted GDNF. However, this may be technically very demanding. Alternatively, the authors could examine GDNF transcripts. Altered GDNF transcript/expression/release is more likely to cause damage to the dopaminergic neurons. As there is an elevated level of Gli1 transcripts in a smaller number of ciliated cells (cholinergic) in R1441C mice, it raises concerns about how much of DA loss might happen when there is an elevated level of Gli1. Addressing the question on GDNF could solve the issue.

We fail to detect loss of total GDNF transcripts in 5 month G2019S striatal cholinergic neurons but we see a shift in GDNF expression to non-ciliated cells, likely due to loss of cilia-dependent Gli3 repression of GDNF transcription (see above). Future experiments will explore this further in older animals that we do not currently have access to.

It would really build the study if you could examine GDNF expression, as is done in figure 6B of the Gonzalez-Reyes et al., (2012) manuscript. I do understand if this is too technically demanding. Alternatively, one could examine GDNF transcripts.

Note that the Gonzalez-Reyes study measured TOTAL striatal GDNF and the ventral striatum contributes significantly to this. We did try transcript detection and only see dysregulation in the mice available to us for this analysis.

Are there humans with PPM1H variants? NCBI's Clinvar database is a very good site to explore this.

At present we are not aware of any PD patients with PPM1H mutations. There are patients with PPM1M mutations, and we are exploring this further. Lack of mutation could be due to functional redundancy or embryonic lethality.

"Rab12 S105 phosphorylation also decreased in wild type (Figure 4C) and mutant (Figure 4A,B) brain (Figure 4). However, levels of pRab10 seemed unchanged (Figure 4A,B)". I don't really understand this seeming discrepancy. I just wonder if there is anything new that could be added.

Several other groups have also reported this (cited above). Remember that cilia are completely normal in many brain regions and changes in pRab10 in the rare cholinergic neurons of the dorsal striatum would not be detected in a whole brain western blot. Sadly we cannot yet stain the cells directly as the phosphoRab10 antibody doesn’t work.

Technical problems:1) Line 65, Dhekne et al., 2021 should be 2018?

2021 is the correct reference that we forgot to include in the ref. List. Thanks for catching this!

2) Figure 6 D, statistical significance is required. Or, the authors mean that there is an increasing trend but not statistically significant.

We mean there is an increasing trend--Gli1 signal is only 1 or 2 dots and the animal to animal variability adds to our challenge at reaching significance. We have corrected the text accordingly.

3) Figure 8 C, representative images would help if possible.

Representative images were presented in (old) Figure 8A but we added many more as requested (New Figure 9).

Reviewer #3:Major points:The LRRK2 inhibitor experiment in the PPM1H^+^/- mice in Figure 5 is confusing- why is this done such a short time beforehand here when in other experiments in Figure 4 the animals were fed the inhibitor for 2 weeks.

We apologize for any confusion. We do not have large numbers of mutant animals. We were not able to feed the PPM1H mutant mice for two weeks with inhibitor; the two hour feeding was used to validate the pRab10 staining and the blots serve to document the genotypes. PPM1H^-/-^ mutant animals (now added here) have higher levels of pRab10 than LRRK2 mutant tissues and in this case we do see MLi-2 sensitivity at 2 hours in whole brain. This is now a supplemental figure and we have clarified the text.

It's also confusing that the text says "they strongly validate a link between LRRK2-Rab phosphorylation and ciliogenesis in specific brain cell types", however the inhibitor experiment doesn't show much change in RAB10 phosphorylation in the PPM1H mice +/- the inhibitor or even the PPM1H mice relative to WT.

The text refers to the single cell phenotype analysis that is not captured in a whole brain western blot. The statement is based on PPM1H specificity for Rab GTPases changing ciliation in a rare neuronal subpopulation.

Some quantification here is necessary to better investigate these changes. And some work could also be done to better clarify the point the authors are trying to make here. More generally, the connection between Rab10 phosphorylation in vivo and the reduction in cilia frequency seems unclear. Is phosphorylation of RAB10 increased in the brain of the R1441C mice compared with WT? Only the knockin mice are shown, and I don't see this in the previous paper.

Please see discussion above. Whole brain was analyzed by western blot and remember that the cholinergic interneurons that we study in the striatum represent only ~5% of the total cells present in that region; pRab10 changes specifically in those cells cannot yet be monitored. Thus, current reagents make it impossible to provide a direct connection between Rab phosphorylation and ciliation in the cholinergic neurons of the striatum. However, in primary cultured astrocytes we see a clear correlation and this is also true for every other cultured cell type analyzed including patient derived G2019S IPs cells. Future work will try to carry out mass spec determinations of pRab10 in small numbers of cholinergic striatal neurons. That a Rab-specific phosphatase mutant has the same cilia loss phenotype as the hyperactive kinase supports the involvement of a phosphoRab.

The Hh signaling data also needs some development and clarification. For the cholinergic neurons- are the overall levels of Gli1 affected? Could this be measured by RNAseq?

See above.

Another way to look at Hh signaling in the cholinergic neurons would be to look at Hh pathway components in cilia. SMO antibodies in particular often work well even in tissue. The authors could test whether SMO localization or intensity in the cilia of the cholinergic neurons is affected which would more directly address whether the cilia that are present are altered in their function as the authors posit.

This is an important experiment for future work. To date our smoothened antibody from Rajat Rohatgi (considered among the best in the field) does not work well in this brain region at this resolution.

The authors also state that reduced numbers of cilia in astrocytes of the R1441C mice leads to an overall reduction in Hh signaling in these cells. I understand they are inferring this from the fact that there are fewer ciliated astrocytes mice but it should also be shown directly.

We have presented the recalculation of the data to show this directly (New Figure 8 C-E; New Figure 7E).